# ATTACKING FOR INSPECTION AND INSTRUCTION: DE-BIASING SELF-EXPLAINING TEXT CLASSIFICATION

## ABSTRACT

*eXplainable Artificial Intelligence* (XAI) techniques are indispensable for increasing the transparency of deep learning models. Such transparency facilitates a deeper human comprehension of the model's fairness, security, robustness, among other attributes, leading to heightened trust in the model's decisions. An important line of research in the field of NLP involves self-explanation using a cooperative game, where a generator selects a semantically consistent subset of the input as the explanation, and a subsequent predictor makes predictions based on the selected subset. In this paper, we first uncover a potential caveat: such a cooperative game could unintentionally introduce a sampling bias between the explanation and the target prediction label. Specifically, the generator might inadvertently create an incorrect correlation between the selected explanation and the label, even when they are semantically unrelated in the original dataset. Subsequently, we elucidate the origins of this bias using both detailed analysis and empirical evidence. Our findings suggest a direction for inspecting this bias through attacking, and we introduce an adversarial game as a practical solution. Experiments [1] on two widely used real-world benchmarks show the effectiveness of the proposed method.

## 1 INTRODUCTION

With the remarkable success of deep learning across various applications, concerns about model interpretability are intensifying. Delving into the theory and techniques of interpretable machine learning frameworks is crucial for addressing a plethora of challenges. For example, XAI techniques can assist in detecting model discrimination (fairness) (Pradhan et al., 2022), pinpointing backdoor attacks (security) (Li et al., 2022), and uncovering potential failure cases (robustness) (Chen et al., 2022), among other issues. Generally, two primary properties are sought in an explanation method: faithfulness and plausibility (Lipton, 2018; Chan et al., 2022). An explanation is considered faithful if it genuinely reflects the model's behavior, and an explanation is deemed plausible if it aligns with human understanding.

Although there have been various methods to generate post-hoc explanations that may appear plausible, they may not faithfully represent an agent's decision, because the process of generating explanations is trained separately from the model's predictions (Lipton, 2018). In certain cases, prioritizing faithfulness over plausibility becomes essential in neural network explanations, especially when these networks are used in vital decision-making processes, as faithfulness directly influences the reliability of the explanations. Unlike post-hoc methods, ante-hoc (or self-explaining) techniques generally provide higher levels of transparency (Lipton, 2018) and faithfulness (Yu et al., 2021), as the prediction is derived directly from the explanation.

In this study, our primary focus is on investigating a general model-agnostic self-explaining framework called Rationalizing Neural Predictions (RNP, also known as rationalization) (Lei et al., 2016), which with its variants has become one of the mainstream methods to facilitate the interpretability of NLP models (Yu et al., 2019; Antognini et al., 2021; Yu et al., 2021; Liu et al., 2022; 2023a;b), and also holds the potential to be applied to image classification (Yuan et al., 2022) and graph neural networks (Luo et al., 2020). RNP utilizes a cooperative game involving a generator and a predictor, in which the generator discerns a human-interpretable subset $Z$ from the input $X$, known as the rationale. This rationale $Z$ is subsequently sent to the following predictor for prediction, as illustrated

---

[1]The code will be confidentially shared with the reviewers during the rebuttal phase.

service is excellent and very clean. good location being close to all destinations in town .

**X**

Generator

service is excellent ~~and very clean.~~ good location being close to all destinations in town .

**Z**

Predictor

[0.1, 0.9]
(Service)

$\hat{Y}$

$min_{\theta_p, \theta_g} H(Y, \hat{Y})$

$Y$ [0, 1]

Figure 1: The standard rationalization framework RNP. $X, Z, \hat{Y}, Y$ represent the input, the selected rationale candidate, the prediction, and the classification label, respectively. $\theta_g, \theta_p$ are the parameters of the generator and the predictor.

in Figure 1. Through cooperative training, the generator and predictor work collaboratively to optimize prediction accuracy. A significant benefit of RNP-based approaches is their ability to certify exclusion, guaranteeing that any unselected input components do not affect the prediction (Yu et al., 2021). This property ensures faithfulness and allows the focus to be on plausibility.

However, the two-stage method of RNP, which is based on a cooperative game, can sometimes lead to a sampling bias that causes plausibility issues including two well-known problems named degeneration[2] (Yu et al., 2019) and irrationality[3] (Zheng et al., 2022). Specifically, the generator $g$ might select rationales that include trivial patterns semantically unrelated to the actual classification labels, and the predictor $p$ then treats these trivial patterns as indicative features for classification. For instance, from a positive input $X^1$ with a label 1, the generator selects a rationale $Z$ that includes the pattern ".", and subsequently the predictor considers the presence of "." as an indicative feature for positive classification. Clearly, in this case, the sampling bias leads to the selection of a semantically irrelevant trivial pattern as an explanation. This results in an explanation that lacks meaningful content, thereby rendering it implausible to human interpreters.

Firstly, in order to inspect and identify this issue, we propose an enhanced method for RNP based on an adversarial game. We introduced an attack generator $g_a$. Figure 2 shows an example of how the attacker works. The optimization objective of $g_a$ is to select an attack rationale $Z_A$ from input such that, when $Z_A$ is fed into the same predictor $p$, it yields a prediction label flipped from its original label. Continuing the previous example, the generator $g$ selects the "." from a positive input $X^1$ with label 1 as $Z$. Consequently, the predictor $p$ learns to treat the presence of "." in $Z$ as an indicative feature for positive classification. On the other hand, the goal of $g_a$ is to se-

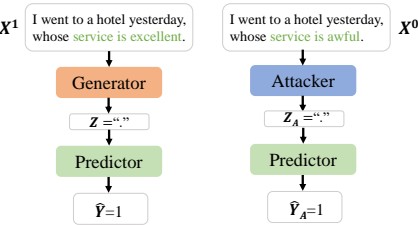

Figure 2: An example of how the attacker works. $X^1, X^0$ represent positive and negative texts, respectively.

lect an attack rationale $Z_A$ from a negative input $X^0$ with a label 0 in such a way that, when $Z_A$ is fed to the same predictor $p$, the prediction result flips from its original label 0 to 1. Achieving this objective is straightforward: $g_a$ simply needs to mimic $g$ by selecting "." as $Z_A$. This suggests that if $g$ identifies $Z$ from $X^1$ as a trivial pattern also present in $X^0$, then $g_a$ can effortlessly select $Z_A = Z$ from $X^0$, leading to an easy flip of the prediction label of $Z_A$ to 1 in predictor $p$. On the other hand, if $Z$ is a genuine positive rationale unique to $X^1$ and the predictor $p$ classifies it correctly, then $g_a$ would be unable to find a positive rationale from the negative input $X^0$. As a result, it becomes difficult for the predictor $p$ to flip $Z_A$'s label from 0 to 1. Therefore, we can leverage the attack generator $g_a$ to assist in inspecting and identifying sampling bias. If $g_a$ can easily find a $Z_A$ that flips its predicted label in predictor $p$ from its actual label, it indicates the presence of semantically unrelated trivial patterns in $Z$.

To further address this issue, we propose a method to instruct the game on better debiasing. As illustrated by the previous example, when there is a sampling bias issue, the attack generator $g_a$ surely selects a $Z_A$ that is a trivial pattern lacking semantic significance. For a reasonable predictor $p$ that can accurately classify the real rationale, $Z_A$ is akin to noise, and its classification result should be random and not biased towards any label. Therefore, we introduce a constraint on the predictor $p$ to guide it, ensuring that the classification result for $Z_A$ remains as random as possible.

---

[2]The definition of degeneration is non-trivial, and we will discuss this problem in Section 2.

[3]Irrationality means the selected rationale candidates may contain both trivial patterns and the real informative rationales, but the predictor makes the prediction according to the trivial patterns. And the problem is that we do not know which part the predictor actually uses. Note that Zheng et al. (2022) do not provide a solution to this problem.

This constraint serves as an ongoing guidance to adjust and correct the behavior of predictor $p$. An improved predictor $p$ can, in turn, better instruct and guide the updates for the generator $g$.

In summary, our contributions lies in the following aspects:

- To the best of our knowledge, we are the first to identify the sampling bias in self-explaining rationalization, which is overlooked by previous research.

- We design an attacker to both inspect whether the predictor has learnt from the bias and instruct the predictor not to learn from the bias. If the predictor learns from the bias, the instruction objective will penalize it, such that the influence of sampling bias is alleviated.

- We design various experiments to verify the existence of sampling bias, the effectiveness of the inspection, and the effectiveness of the instruction. Besides, the attack based inspection and instruction is model-agnostic, so we conduct is on top of both the vanilla RNP and an advance method FR (Liu et al., 2022), and all get improved performance.

## 2 RELATED WORK

**Rationalization**. The base cooperative framework of rationalization named RNP (Lei et al., 2016) is flexible and offers a unique advantage: certification of exclusion, which means any unselected input is guaranteed to have no contribution to prediction (Yu et al., 2021). Based on this cooperative framework, many methods have been proposed to improve RNP from different aspects. Bao et al. (2018) used Gumbel-softmax to do the reparameterization for binarized selection. Bastings et al. (2019) replaced the Bernoulli sampling distributions with rectified Kumaraswamy distributions. Jain et al. (2020) disconnected the training regimes of the generator and predictor networks using a saliency threshold. Paranjape et al. (2020) imposed a discrete bottleneck objective to balance the task performance and the rationale length. Zheng et al. (2022) called for more rigorous evaluations of rationalization models. Fernandes et al. (2022) leveraged meta-learning techniques to improve the quality of the explanations. Havrylov et al. (2019) cooperatively trained the models with standard continuous and discrete optimisation schemes. Hase et al. (2020) explored better metrics for the explanations. Rajagopal et al. (2021) used phrase-based concepts to conduct a self-explaining model. Other methods like data augmentation with pretrained models (Plyler et al., 2021), training with human-annotated rationales (Chan et al., 2022), injecting noise to the selected rationales (Storek et al., 2023), have also been tried. These methods are orthogonal to our research.

Another series of papers that are most related to our work are those discussing the degeneration problem. Degeneration means that, the predictor is too powerful to recognize any trivial patterns that are distinguishable in rationales with opposite labels. As a result, the generator may collude with the predictor to select the trivial patterns rather than the true semantics as the rationales (Yu et al., 2019). This problem is very similar to what we discuss. And the sampling bias we discuss can be seen as a reason why degeneration happens. Previous methods seek to regularize the predictor using supplementary modules which have access to the information of the full text (Yu et al., 2019; Huang et al., 2021; Yu et al., 2021; Liu et al., 2022; Yue et al., 2022) such that the generator and the predictor will not collude to uninformative rationales. 3PLAYER (Yu et al., 2019) takes the unselected text $Z^c$ into consideration by inputting it to a supplementary predictor *Predictor$^c$*. DMR (Huang et al., 2021) tries to align the distributions of rationale with the full input text in both the output space and feature space. A2R (Yu et al., 2021) endows the predictor with the information of full text by introducing a soft rationale. FR (Liu et al., 2022) folds the two players to regularize each other by sharing a unified encoder. These methods are most related to our work. However, these methods only try to **fix** the degeneration problem, while where the problem derives is not well discussed. Sometimes they can still fail. For example, (Zheng et al., 2022) argued with both philosophical perspectives and empirical evidence that the degeneration problem is much complex than we used to think and some of the above methods cannot promise no-degeneration. To the best of our knowledge, we are the first one to consider it as a kind of sampling bias.

**Generative Explanation with Large Language Models**. Generative explanation is a research line that is close but orthogonal to our research. With the great success of large language models (LLMs), a new research line for explanation is chain-of-thought. By generating (in contrast to selecting) intermediate reasoning steps before inferring the answer, the reasoning steps can be seen as a kind of explanation. The intriguing technique is called chain-of-thought (CoT) reasoning (Wei et al.,

2022). However, LLMs sometimes exhibit unpredictable failure modes (Kıcıman et al., 2023) or hallucination reasoning (Ji et al., 2023), making this kind of generative explanation not trustworthy enough in some high-stakes scenarios. Also, some recent research finds that LLMs are not good at extractive tasks (Qin et al., 2023; Li et al., 2023; Ye et al., 2023).

## 3 PROBLEM DEFINITION

For the sake of exposition, let us take the example of binary sentiment classification. Generalization to multi-class classification is in Appendix A.2. We have a dataset $\mathcal{D}$, which consists of a set of $(X, Y)$ pairs and can be seen as a collection of samples drawn from the true data distribution $P(X, Y)$. $X = X_{1:l}$ is the input text sequence with a length of $l$, and $Y$ is the discrete class label. By enumerating $X$, we can get $P(Y|X)$, which is the distribution that a normal non-interpretable classifier working on $\mathcal{D}$ needs to approximate. Self-explaining rationalization consists of a generator $f_g(\cdot)$ (or $g$ for conciseness) and a predictor $f_p(\cdot)$, with $\theta_g, \theta_p$ being their parameters, respectively.

In self-explaining rationalization, for $(X, Y) \in \mathcal{D}$, the generator first outputs a sequence of binary mask $M = f_g(X) = M_{1:l} \in \{0, 1\}^l$ (in practice, the generator first outputs a Bernoulli distribution for each token and the mask for each token is independently sampled using gumbel-softmax). Then, it forms the rationale candidate $Z$ by the element-wise product of $X$ and $M$:

$$Z = M \odot X = [M_1 X_1, \cdots, M_l X_l]. \tag{1}$$

To simplify the notation, we denote $f_g(X)$ as $Z$ in the following sections, i.e., $f_g(X) = Z$.

We consider that $X$ consists of a set of variables $\{T_1, \cdots, T_n, S\}$, where $S$ denotes real rationale for corresponding sentiment label $Y$, and $T_1, \cdots, T_n$ are some **t**rivial patterns independent with $Y$. And we select one of $\{T_1, \cdots, T_n, S\}$ to be $Z$. It is worth noting that $Z$ is not a separate variable but a proxy for any variable within $X$. Till now, we get a set of $(Z, Y)$ pairs denoted as $\mathcal{D}_{\mathcal{Z}}$. Vanilla RNP simply thinks $\mathcal{D}_{\mathcal{Z}}$ is collected from $P(Z, Y)$. By enumerating $Z$ in $\mathcal{D}_{\mathcal{Z}}$, it gets $P(Y|Z)$. Then, RNP attempts to identify the rationale by maximizing the mutual information $I(Y; Z)$:

$$Z^* = \underset{Z \in \{T_1, \cdots, T_n, S\}}{\arg\max} I(Y; Z) = \underset{Z \in \{T_1, \cdots, T_n, S\}}{\arg\max} (H(Y) - H(Y|Z)) = \underset{Z \in \{T_1, \cdots, T_n, S\}}{\arg\min} H(Y|Z). \tag{2}$$

In practice, the entropy $H(Y|Z)$ is commonly approximated by the minimum cross-entropy $\min_{\theta_p} H_c(Y, \hat{Y}|Z)$, with $\hat{Y} = f_p(Z)$ representing the output of the predictor. It is essential to note that the minimum cross-entropy is equal to the entropy (please refer to Appendix B.1). Replacing $Z$ with $f_g(X)$, the explainer and the predictor are trained cooperatively:

$$\min_{\theta_g, \theta_p} H_c(Y, f_p(f_g(X))|f_g(X)), \ s.t., \ (X, Y) \sim \mathcal{D}. \tag{3}$$

**Compactness and coherence**. To make the selected rationales human-intelligible, previous methods usually constrains the rationales by compact and coherent regularization terms. In this paper, we use the most widely used constraints provided by Chang et al. (2019):

$$\Omega(M) = \lambda_1 \left| \frac{\|M\|_1}{l} - s \right| + \lambda_2 \sum_{t=2}^{l} |M_t - M_{t-1}|. \tag{4}$$

The first term encourages that the percentage of the tokens being selected as rationales is close to a pre-defined level $s$. The second term encourages the rationales to be coherent. We adopt both compactness and coherence regularizers to the generator to make the rationales human-intelligible. We apply a compactness regularizer term to the attacker to make the attack rationale more similar to the original rationale, thus making it easier to deceive the predictor. However, we do not employ a coherence regularizer on it because we think trivial patterns are often discontinuous.

## 4 METHOD AND MOTIVATION

### 4.1 METHOD

The architecture of our method is shown in Figure 3. The overall objective of our model is

$$\text{gen\& pred}: \min_{\theta_g,\theta_p} H_c(Y, f_p(f_g(X))|f_g(X)) + \min_{\theta_p} H_c([0.5, 0.5], f_p(f_a(X))|f_a(X)), \quad (5)$$

$$\text{attacker}: \min_{\theta_a} H_c(1 - Y, f_p(f_a(X))|f_a(X)), \quad (6)$$

where $f_p(\cdot), f_g(\cdot), f_a(\cdot)$ represent the predictor, the generator, and the attacker. And $\theta_p, \theta_g, \theta_a$ are their parameters. During training, Equations (5) and (6) are alternated. The practical implementation details with Pytorch are in Appendix A.1. The overall mechanism of the model is as follows: Equation (6) inspects trivial patterns ($f_a(X)$) from $X$. The second term of Equation (5) is the instruction that prevents the predictor from learning the trivial patterns by classifying them as random noise. A well instructed predictor is then able to give good feedback to the generator's selection. And the first term of Equation (5) is the normal RNP. The reason why the attacker constructed in this manner can detect trivial patterns will be explained in detail in Section 4.2.

### 4.2 MOTIVATION

**Notation.** We denote $X^1$ and $X^0$ as input texts with label $Y = 1$ and $Y = 0$, respectively. $Z$ and $Z_A$ represent the rationale candidates selected by the generator and the attacker, respectively. Note that they are not separate variables but a proxy for any variables within $X$. Sometimes we use $Z$ and the variable represented by $Z$ interchangeably. $T$ is a proxy for any variables within $\{T_1, \cdots, T_n\}$. Lowercase letters denote the values of variables.

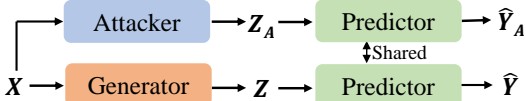

Figure 3: Our proposed method. $X, Z, \hat{Y}, Y$ represent the input, the selected rationale candidate, the prediction and the class label, respectively.

**How does the sampling bias come into being?** Although considering $\mathcal{D}_{\mathcal{Z}}$ as an approximation of $P(Z,Y)$ seems to be a simple and practical way and is inherited by all the previous methods, it will sometimes results in some problems. In fact, the sampling process of $Z$ is conditioned on a generator $g$ with specific parameters $\theta_g$. So we can only get $P(Z,Y|g)$ and $P(Y|Z,g)$ rather than $P(Z,Y)$ and $P(Y|Z)$. Note that independent doesn't lead to conditional independent: $Y \perp\!\!\!\perp Z \not\Rightarrow Y \perp\!\!\!\perp Z|g$.

That is to say, some uninformative $Z$ (like those $T_1, \cdots, T_n$) might initially be semantically unrelated to $Y$ and maintain zero mutual information with $Y$, indicating their independence. But sampled by $g$, any trivial patterns may get correlated with $Y$ and get increased mutual information, thus can be used as (incorrect) indicative features for classification by the predictor.

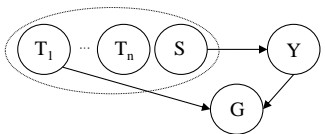

What's more, we find the training process may even enhance the sampling bias further. For example, we consider $T_1$ is selected as $Z$, then the updating of the generator should be $\theta'_g = h(\theta_g, T_1, Y)$ ($h$ denotes the back propagation function), which corresponds to a small local of a causal graph shown in Figure 4. We originally have

Figure 4: A small local of the causal graph for the generator's updating process. The dash cycle means $X$ consists of a set of variables.

$Y \perp\!\!\!\perp T_1$. But in this graph, we have $Y \not\perp\!\!\!\perp T_1|G$. That's to say, any trivial patterns hold the potential to be associated with $Y$ through the influence of the generator.

**Sampling bias can make trivial patterns indicative to the sentiment label and result in a bad predictor.** Consider a situation where $Z = T$ is a trivial pattern independent with $Y$ (i.e., $P(Y = 1|T) = P(Y = 1) = 0.5 = P(Y = 0) = P(Y = 0|T)$ and $T \in \{t_+, t_-\}$). Influenced by the generator $g$, $T = t_+$ might co-occur more frequently with $Y = 1$ and can be viewed as an indicator for the positive class ($T = t_-$ is similar):

$$\begin{cases} P(Y = 1|Z = t_+, g) > P(Y = 1) = 0.5 = P(Y = 0) > P(Y = 0|Z = t_+, g), \\ P(Y = 1|Z = t_-, g) < P(Y = 1) = 0.5 = P(Y = 0) < P(Y = 0|Z = t_-, g). \end{cases} \quad (7)$$

Here is an intuitive toy example. We consider $Z = T$ is a punctuation pattern. $t_+$ represents ".", and $t_-$ represents ",". For example, we have a positive text $X^1$ and a negative text $X^0$. Chances are that

the generator $g$ selects $Z$="." from $X^1$ and $Z$="," from $X^0$. It appears that $P(Y = 1|Z = ".", g)$ and $P(Y = 0|Z = ",", g)$ are very high. As a result, the predictor can just use "." and "," to get a high predictive accuracy, even if the punctuation patterns are semantically unrelated to $Y$ and maintain low mutual information with $Y$ in the original dataset. This part is verified by the experiments in Section 5.1 to some extent.

**Attack as inspection**. Following the above settings for $Z = T$ and $I(Y; T) = 0$, we will show how the trivial patterns learned by the predictor can be inspected. If the attack generator can be constructed in any way (i.e., has infinite expressiveness), then we can also find another attack generator $g_a$ which finds $Z_A$ from $X$, such that

$$\begin{cases} P(Y = 1|Z_A = t_+, g_a) < P(Y = 1) = 0.5 = P(Y = 0) < P(Y = 0|Z_A = t_+, g_a), \\ P(Y = 1|Z_A = t_-, g_a) > P(Y = 1) = 0.5 = P(Y = 0) > P(Y = 0|Z_A = t_-, g_a). \end{cases} \quad (8)$$

Appendix B.2 shows the detailed derivation for the reason why we can find such a $g_a$. Equation (8) means that under condition $g_a$, $T = t_+$ now becomes a negative class indicator, which is exactly the opposite situation under condition $g$. Here is the intuitive understanding of the attack. Corresponding to the punctuation pattern example mentioned above. The generator $g$ selects $Z = "."$ from $X^1$. And the predictor has learnt to predict "." as positive. We can employ an attacker $g_a$ which selects $Z_A = "."$ from $X^0$ (note that the label of $X^0$ is negative) such that $Z_A$ can also be classified as positive. Similarly, the attacker can find $Z_A = ","$ from $X^1$ to be classified as negative. So, the overall objective of the attacker is to select those $Z_A$ that can be classified to the opposite class by the predictor.

Formally, the objective of the attacker is

$$\min_{\theta_a} H_c(1 - Y, f_p(f_a(X))|f_a(X)), \ s.t., \ (X, Y) \sim \mathcal{D}, \quad (9)$$

where $f_a(\cdot)$ is the attacker with $\theta_a$ being its parameters, and $Z_A = f_a(X)$.

In the discussion above, we demonstrated that an attacker can identify uninformative trivial patterns and classify them into the opposite class. Then we begin to instruct the predictor to not learn from the trivial patterns (whether the attacker will select real rationales is discussed later).

**Attack as instruction.** When sampling bias arises, the attack generator $g_a$ consistently chooses a $Z_A$ which is a semantically insignificant trivial pattern. For a competent predictor $p$ that discerns the authentic rationale, $Z_A$ resembles noise, ensuring its classification remains random without any leanings to a specific label. Thus, we introduce an extra instruction to the predictor:

$$\min_{\theta_p} H_c([0.5, 0.5], f_p(Z_A)), \ s.t., \ Z_A = f_a(X), \ (X, Y) \sim \mathcal{D}, \quad (10)$$

where $f_p(\cdot), \theta_p$ denote the predictor and its parameters, respectively. The objective for multi-class classification is in Appendix A.2.

We have discussed the situations where the predictor and the generator overfit to trivial patterns. Under these situations, the attacker will select trivial patterns with $Z_A = f_a(X)$, and thus Equation (10) will instruct the predictor to classify $Z_A$ as noise. The following question is, if the generator and the predictor work well on selecting real rationales, will Equation (10) do harm to the predictor?

**The instruction will not cause harm to a good predictor.** Here we consider $Z = S$, which is the real sentiment rationale based on which the label $Y$ is assigned to $X$. We denote $S = s_+, S = s_-$ as positive and negative sentiments, respectively. If a good predictor learns to use $s_+$ as the positive indicator, it will be hard for the attacker to find $Z_A = s_+$ from $X^0$, since a negative text usually does not have a positive sentiment (the discussion about some counterexamples of this assumption is in Appendix B.3). As a result, the attacker can only select certain neutral patterns as $Z_A$ from $X^0$ to shift $f_p(Z_A)$ away from $0$. Hence Equation (10) still will not cause harm to the predictor.

## 5 EXPERIMENTS

In this section, we name our method **A**ttack to **I**nspection and **I**nstruction (A2I). We first verify that sampling bias can be used for classification (resulting in a poor predictor). Then, we show the effectiveness of our method in dealing with sampling bias using two widely used rationalization benchmarks.

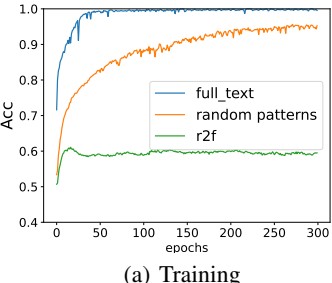 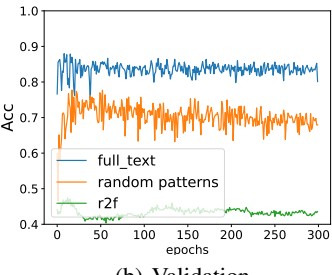



(a) Training        (b) Validation



Figure 5: Experiments on the *Aroma* aspect of the BeerAdvocate dataset: "full text": a predictor trained using the full texts. "random patterns": a predictor trained with randomly selected patterns. "r2f": feeding the random patterns to the predictor that was trained using the full texts.

## 5.1 RANDOMLY SELECTED PATTERNS CAN BE TREATED AS INDICATIVE FEATURES FOR CLASSIFICATION

We present three types of prediction accuracies for the BeerAdvocate dataset: (1) A predictor trained with the full input text. (2) A predictor trained with randomly selected patterns. For the generator, we remove the other objectives and only train it with the sparsity constraints. Specifically, the generator is trained to randomly select $10\%$ of the input text, and the predictor is then trained to classify using these randomly selected texts. (3) We use the randomly selected texts from (2) to feed the predictor trained in (1).

The result for the *Aroma* aspect is shown in Figure 5. From Figure 5(a), we observe that even with the randomly selected patterns (i.e., patterns unlikely to contain real rationales), the predictor can still achieve a very high prediction accuracy (represented by the orange line, approximately $95\%$). This accuracy is close to that of the classifier trained with the full texts. A followed question is: Does this result suggest that the $10\%$ randomly selected patterns already contain enough sentiment inclination for classification? The answer is no. Consider the green line, which represents the outcome when we feed the randomly selected texts to the predictor denoted by the blue line. We observe that the green line indicates a significantly lower accuracy (about $58\%$), implying that the randomly selected patterns contain only minimal sentiment information. Thus, the orange predictor incorrectly treats certain randomly selected trivial patterns as indicative features. Moreover, the orange predictor does not generalize well to the validation set, as depicted in Figure 5(b). This is likely because simple trivial patterns can more easily lead to overfitting (Pagliardini et al., 2023).

## 5.2 EXPERIMENTS ON STANDARD BENCHMARKS

### 5.2.1 SETTINGS

**Baselines**. The primary baseline for direct comparison is the original cooperative rationalization framework, RNP (Lei et al., 2016). This helps us concentrate on our claims rather than on potential unknown mechanisms. To demonstrate the competitiveness of our method, we also include two recently published representative models: Inter_RAT (Yue et al., 2023) and FR (Liu et al., 2022). Both of them have been discussed in Section 2.

**Datasets**. Following Liu et al. (2022), we examine two widely-used datasets for rationalization tasks. ***BeerAdvocate*** (McAuley et al., 2012) is a dataset for multi-aspect sentiment prediction related to beer reviews. Following FR, we use the subsets decorrelated by Lei et al. (2016). ***HotelReview*** (Wang et al., 2010) is another multi-aspect sentiment prediction dataset focusing on hotel reviews. This dataset includes reviews from three aspects: location, cleanliness, and service. Both datasets feature human-annotated rationales in the annotation (test) set. We preprocess both datasets in the same manner as FR (Liu et al., 2022) to ensure a fair comparison; more details are in Appendix A.3.

**Implementation details**. Experiments in recent works show that it is still a challenging task to fine-tune over-parameterized pretrained language models like BERT (Devlin et al., 2019) on the RNP cooperative framework (Chen et al., 2022; Liu et al., 2022; Zhang et al., 2023). The detailed dis-

Table 1: Results on *BeerAdvocate*. Each aspect is trained independently

| Methods | Appearance | | | | | Aroma | | | | | Palate | | | | |
|---|---|---|---|---|---|---|---|---|---|---|---|---|---|---|---|
| | S | Acc | P | R | F1 | S | Acc | P | R | F1 | S | Acc | P | R | F1 |
| Comparison with vanilla RNP | | | | | | | | | | | | | | | |
| RNP | 10.1 | 79.7 | 69.3 | 37.6 | 48.8 | 10.0 | 82.9 | 81.3 | 52.4 | 63.7 | 9.3 | 84.7 | 68.6 | 51.3 | 58.7 |
| RNP+A2I | 10.8 | 82.8 | 78.3 | 45.8 | **57.8** | 9.8 | 86.3 | 86.0 | 54.3 | **66.6** | 10.9 | 86.6 | 66.3 | 58.2 | **62.0** |
| RNP | 19.8 | 86.3 | 69.8 | 74.6 | 72.1 | 20.7 | 84.5 | 43.6 | 58.1 | 49.8 | 20.1 | 82.6 | 47.6 | 77.0 | 58.8 |
| RNP+A2I | 20.0 | 87.7 | 73.3 | 79.4 | **76.2** | 19.5 | 85.4 | 49.0 | 61.4 | **54.5** | 19.4 | 86.6 | 49.0 | 76.4 | **59.7** |
| RNP | 30.4 | 84.3 | 52.9 | 86.7 | 65.7 | 30.7 | 81.8 | 39.2 | 77.2 | 52.0 | 30.1 | 87.1 | 29.3 | 71.0 | 41.5 |
| RNP+A2I | 29.9 | 85.2 | 59.3 | 95.9 | **73.3** | 27.8 | 87.3 | 44.5 | 79.3 | **57.0** | 30.5 | 87.1 | 30.8 | 75.5 | **43.7** |
| Comparison with advanced methods | | | | | | | | | | | | | | | |
| Inter_RAT | 13.2 | - | 50.0 | 35.7 | 41.6 | 13.8 | - | 64.0 | 56.9 | 60.2 | 13.0 | - | 47.2 | 49.3 | 48.2 |
| FR | 11.0 | 82.2 | 68.0 | 40.5 | 50.8 | 9.4 | 86.7 | 85.3 | 51.5 | 64.2 | 9.4 | 84.5 | 70.1 | 52.8 | 60.2 |
| FR+A2I | 11.3 | 84.6 | 76.0 | 46.5 | **57.7** | 10.0 | 86.9 | 85.7 | 54.8 | **66.9** | 9.7 | 84.8 | 71.4 | 55.8 | **62.6** |
| Inter_RAT | 20.2 | - | 45.8 | 50.4 | 48.0 | 22.0 | - | 47.2 | 67.3 | 55.5 | 20.2 | - | 39.9 | 64.9 | 49.4 |
| FR | 19.7 | 87.7 | 77.7 | 82.8 | 80.2 | 20.5 | 90.5 | 61.1 | 80.3 | 69.4 | 19.8 | 86.0 | 42.1 | 67.0 | 51.7 |
| FR+A2I | 19.8 | 88.7 | 80.0 | 85.6 | **82.7** | 19.4 | 89.7 | 64.2 | 80.0 | **71.2** | 19.2 | 86.0 | 44.2 | 68.2 | **53.7** |
| Inter_RAT | 28.3 | - | 48.6 | 74.9 | 59.0 | 31.5 | - | 37.4 | 76.2 | 50.2 | 29.2 | - | 29.7 | 69.7 | 41.7 |
| FR | 30.0 | 90.9 | 58.5 | 94.6 | 72.3 | 31.0 | 83.2 | 40.0 | 79.4 | 53.2 | 29.3 | 84.8 | 28.5 | 67.2 | 40.1 |
| FR+A2I | 28.8 | 89.7 | 61.3 | 95.3 | **74.6** | 30.9 | 83.2 | 41.4 | 82.2 | **55.1** | 29.1 | 85.1 | 31.6 | 73.8 | **44.2** |

Table 2: Results on *HotelReview*. Each aspect is trained independently.

| Methods | Location | | | | | Service | | | | | Cleanliness | | | | |
|---|---|---|---|---|---|---|---|---|---|---|---|---|---|---|---|
| | S | Acc | P | R | F1 | S | Acc | P | R | F1 | S | Acc | P | R | F1 |
| Comparison with vanilla RNP | | | | | | | | | | | | | | | |
| RNP | 8.8 | 97.5 | 46.2 | 48.2 | 47.1 | 11.0 | 97.5 | 34.2 | 32.9 | 33.5 | 10.5 | 96.0 | 29.1 | 34.6 | 31.6 |
| RNP+A2I | 9.0 | 97.5 | 50.2 | 53.4 | **51.7** | 11.6 | 97.0 | 46.8 | 47.4 | **47.1** | 9.7 | 96.5 | 34.7 | 38.2 | **36.4** |
| Comparison with advanced methods | | | | | | | | | | | | | | | |
| Inter_RAT | 11.0 | 95.5 | 34.7 | 44.8 | 39.1 | 12.5 | 98.5 | 35.4 | 39.1 | 37.2 | 9.6 | 97.0 | 33.4 | 36.7 | 34.9 |
| FR | 9.0 | 93.5 | 55.5 | 58.9 | 57.1 | 11.5 | 94.5 | 44.8 | 44.7 | 44.8 | 11.0 | 96.0 | 34.9 | 43.4 | 38.7 |
| FR+A2I | 9.9 | 94.0 | 53.2 | 62.1 | **57.3** | 11.5 | 97.0 | 47.7 | 47.7 | **47.7** | 10.8 | 95.5 | 35.9 | 43.7 | **39.4** |

cussion about BERT is in Appendix A.4. To avoid being influenced by unknown issues and result in potential unfairness in comparisons, we take the same setting as Inter_RAT and FR do: We use one-layer 200-dimension bi-directional gated recurrent units (GRUs) (Cho et al., 2014) followed by one linear layer for each of the players, and the word embedding is 100-dimension Glove (Pennington et al., 2014). The optimizer is Adam (Kingma & Ba, 2015). The reparameterization trick for binarized sampling is Gumbel-softmax (Jang et al., 2017; Bao et al., 2018), which is also the same as the baseline FR. All of the models are trained on a RTX3090 GPU.

**Metrics**. The sampling bias makes the prediction performance not a good metric for the models' effectiveness. Following Inter_RAT and FR, we mainly focus on the rationale quality, which is measured by the overlap between model-selected tokens and human-annotated rationales. The terms $P, R, F1$ denote precision, recall, and $F1$ score respectively. The term $S$ represents the average sparsity of the selected rationales, that is, the percentage of selected tokens in relation to the full text. $Acc$ stands for the predictive accuracy on the test set.

### 5.2.2 RESULTS

**Rationale quality**. Table 1 shows the results in the three aspects of the *BeerAdvocate* dataset. Since each aspect is trained independently, they can each be considered distinct datasets to some extent. Given that the sparsity of human-annotated rationales varies significantly across different aspects, we follow Inter_RAT to set there different levels: $10\%, 20\%, 30\%$, by adjusting $s$ in Equation (4). Initially, we conduct our attacking inspection on top of the vanilla RNP to validate our claims and demonstrate the efficacy of our proposed method. Across all nine settings, we observe a significant improvement over the vanilla RNP in terms of F1 score. Notably, the highest increase reaches up to $9.0\%$ (the *Appearance* aspect with $S \approx 10$), underscoring the robust effectiveness of our method.

Our A2I is model-agnostic; therefore, we further apply it on top of the advanced method, FR, to demonstrate our competitiveness. Two observations emerge from the results. Firstly, neither Inter_RAT nor FR consistently excels across all aspects. While FR performs impressively on the *Appearance* and *Aroma* aspects, it does not surpass RNP on the *Palate* aspect. However, when our attacking inspection is incorporated, the performance of both RNP and FR consistently improves.

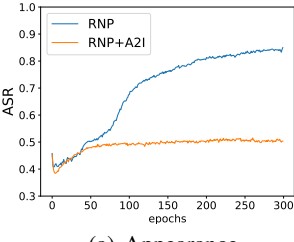 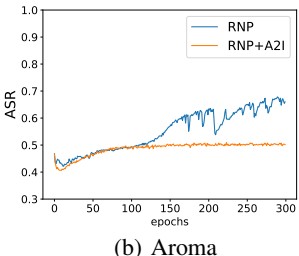 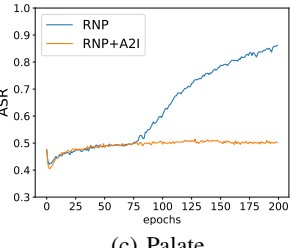

(a) Appearance            (b) Aroma            (c) Palate

Figure 6: Attack success rate (ASR) on the three aspects of *BeerAdvocate* dataset. The rationale sparsity is about $20\%$. More results for sparsity being $10\%$ and $30\%$ is in Appendix A.5.

We observe a significant improvement in FR's performance (up to $6.9\%$ on the *Appearance* aspect with $S \approx 10$) when our A2I is layered atop it, highlighting the competitiveness of our method. Besides the widely-used *BeerAdvocate* dataset, we also follow FR's lead and incorporate the *Hotel-Review* dataset as supplementary material. The results are detailed in Table 2. We consistently achieve strong performance across the three aspects of this dataset.

**Attack Success Rate (ASR)**. To more effectively demonstrate the capabilities of our attacking inspection, we present the attack success rates for both RNP and our RNP+A2I. This experiment aims to address two key questions: 1) Can the attacker truly identify the trivial patterns recognized by the predictor? 2) Can the inspection really prevent the predictor from adopting the trivial patterns? ASR is a metric commonly employed in the realm of security. Given a pair $(X, Y)$, if $f_p(f_a(X)) = 1 - Y$, indicating a label inversion, we deem the attack successful. ASR serves as an indicator of both an attack method's efficacy and a model's resilience against such attacks. A high ASR signifies the effectiveness of an attack method, while a low ASR denotes model robustness.

The results for the three aspects of *BeerAdvocate* are displayed in Figure 6. The rationale sparsity is set at approximately $20\%$. More results with different sparsity can be found in Appendix A.5. Regarding the first question, "Can the attacker truly identify the trivial patterns learned by the predictor?", the blue lines offer insight. As opposed to RNP+A2I, the blue lines depict models where we omit the objective Equation (10) (specifically, the instruction loss) from Equation (5). This means that while RNP is trained as usual, an attacker is also being trained concurrently. The prominence of the blue lines demonstrates that the attacker achieves a remarkably high ASR. This indicates that the predictor in RNP does internalize some trivial patterns, and the attacker successfully identifies them, underscoring the potency of the attack. For the second question, "Can the inspection effectively deter the predictor from adopting trivial patterns?", we can look to the orange lines. The ASR values hover around $50\%$, which is close to random classification. This suggests that the predictor actively avoids learning from the trivial patterns, highlighting the efficacy of the instruction.

## 6   CONCLUSION, LIMITATIONS AND FUTURE WORK

In this paper, we first identify that previous rationalization methods that select rationales through maximum mutual information criterion face a sampling bias problem that arises from neglecting the influence of the generator on $P(Y|Z)$. We design an attacker to first inspect the bias and then give the instruction to prevent the predictor from adopting this bias. The potential impact is twofold. First, to the best of our knowledge, this is the first time to discuss that the sampled $(Z, Y)$ pairs may not represent the distribution $P(Z, Y)$, which could serve as a reminder for future researchers to be more cautious when making assumptions. Second, the attack-based inspection and instruction are model-agnostic and hold the potential to be combined with future research.

One limitation is that our analysis focuses on the classification task, and further research is needed to extend it to broader domains. Another limitation is that the obstacles in utilizing powerful pre-trained language models under the rationalization framework remain mysterious. Although we have discussed some possible reasons that may prevent pretrained models from achieving good performance, we agree that formally investigating this problem is important. However, it is somewhat beyond the scope of this paper, and we leave it as the future work.

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

## A  MORE RESULTS

### A.1  IMPLEMENTATION DETAILS OF EQUATION (6) AND (5)

For a batch of $(X, Y)$, we first send $X$ to both the generator and the predictor and get $Z, Z_A$:

$$Z = f_g(X)$$
$$Z_A = f_a(X). \tag{11}$$

Then, we get a copy of $Z_A$ with the pytorch function "torch.detach()":

$$Z'_A = \text{torch.detach}(Z_A). \tag{12}$$

Then we get $\hat{Y}$ and $\hat{Y}'_A$:

$$\hat{Y} = f_p(Z)$$
$$\hat{Y}'_A = f_p(Z'_A) \tag{13}$$

Then we can update the generator and the predictor with

$$\min_{\theta_g, \theta_p} H_c(Y, \hat{Y}) + \min_{\theta_p} H_c([0.5, 0.5], \hat{Y}'_A) \tag{14}$$

Note that this updating process will not influence the attacker, since we have used "torch.detach()" for $Z_A$.

Then, we fix the parameters of the generator and the predictor, and only update the attacker. We get $\hat{Y}_A$ with

$$\hat{Y}_A = f_p(Z_A). \tag{15}$$

Then, we update the attacker with

$$\min_{\theta_a} H_c(1 - Y, \hat{Y}_A). \tag{16}$$

Then, we get into the next round to update the generator and the predictor again.

### A.2  GENERALIZATION TO MULTI-CLASS CLASSIFICATION

While we primarily focus on binary classification for the sake of exposition brevity, the method can easily generalize to multi-class classification.

All we need to do is modifying Equation (9) and (10). Equation (9) should be modified to

$$\min_{\theta_a}[\min_{Y'} H_c(Y', f_p(f_a(X))|f_a(X))], \ s.t., \ (X, Y) \sim \mathcal{D}, \ Y' \neq Y. \tag{17}$$

which means that $Z_A$ can be classified to any classes except $Y$.

And Equation (10) should be modified to

$$\min_{\theta_p} H_c([\frac{1}{n}, \cdots, \frac{1}{n}], f_p(f_a(X))|f_a(X)), \ s.t., \ (X, Y) \sim \mathcal{D}. \tag{18}$$

Note that we need human-annotated rationales to test the model performance, so there is no proper multi-class classification datasets in the field of rationalization. The datasets we select are just the same as our baseline FR.

### A.3  EXPERIMENTAL DETAILS

**BeerAdvocate**. Following Inter_RAT and FR, we consider a classification setting by treating reviews with ratings $\leq 0.4$ as negative and $\geq 0.6$ as positive. Then we randomly select examples from the original training set to construct a balanced set.

Table 3: Statistics of datasets used in this paper

| Datasets | | Train | | Dev | | Annotation | | |
|---|---|---|---|---|---|---|---|---|
| | | Pos | Neg | Pos | Neg | Pos | Neg | Sparsity |
| Beer | Appearance | 16891 | 16891 | 6628 | 2103 | 923 | 13 | 18.5 |
| | Aroma | 15169 | 15169 | 6579 | 2218 | 848 | 29 | 15.6 |
| | Palate | 13652 | 13652 | 6740 | 2000 | 785 | 20 | 12.4 |
| Hotel | Location | 7236 | 7236 | 906 | 906 | 104 | 96 | 8.5 |
| | Service | 50742 | 50742 | 6344 | 6344 | 101 | 99 | 11.5 |
| | Cleanliness | 75049 | 75049 | 9382 | 9382 | 99 | 101 | 8.9 |

**HotelReviews**. Similar to BeerAdvocate, we treat reviews with ratings < 3 as negative and > 3 as positive.

More details are in Table 3. *Pos* and *Neg* denote the number of positive and negative examples in each set. *Sparsity* denotes the average percentage of tokens in human-annotated rationales to the whole texts.

We get the license of BeerAdvocate by sending an email to Julian McAuley. The Hotel Reviews is a public dataset and we get it from https://github.com/kochsnow/distribution-matching-rationality.

Table 4: The F1 scores of models trained with BERT encoder. The results are obtained from (Chen et al., 2022). The dataset is decorrelated *Beer-Appearance*.

| Method | BERT |
|---|---|
| VIB (Paranjape et al., 2020) | 20.5 |
| SPECTRA (Miyato et al., 2018) | 28.6 |

### A.4 THE DISCUSSION ABOUT THE BERT

The results of several recent papers Chen et al. (2022); Liu et al. (2022); Zhang et al. (2023) have shown that the rationalization framework doesn't perform well when combined with pretrained encoders like BERT. Table 4 shows that two advanced methods, VIB and SPECTRA, both perform much worse than the vanilla RNP with GRUs. Table 5 shows the recent method FR also performs very poorly.

We can also refer to Table 1 of a recent paper CR (Zhang et al., 2023), which shows that none of the rationalization methods gets a F1 score higher than $40.0$ (the sparsity is about $10\%$ in CR) on the Beer dataset when they are conducted with BERT. Compared to our RNP-GRU in Table 1, the lowest F1 for the simplest RNP with $S \approx 10.0$ is $48.8$.

Table 5: The F1 scores of models trained with BERT encoder. The results are obtained from (Liu et al., 2022). The dataset is decorrelated *Beer-Appearance*. The rationale sparsity is about $18\%$.

| Method | BERT |
|---|---|
| RNP (Lei et al., 2016) | 14.7 |
| FR (Liu et al., 2022) | 29.8 |

Here are some possible reasons for the poor results with BERT. First, the rationalization framework usually involves many hyperparameters (e.g., the short and coherent regularizers in Equation (4)), and the over-parameterized BERT may be very sensitive to hyperparameter tuning. For example, the Remark 6.1 in (Zhang et al., 2023) shows that a very small change in the learning rate will cause a very different result. Second, the the over-parameterized BERT is too powerful. With BERT, the predictor may be able to make the right prediction with any trivial patterns, thus the generator does not need to select the real rationales.

These reasons may not be true. However, exploring what happens to BERT is somewhat beyond the scope of this paper, and we leave it as the future work. To avoid being influenced by unknown issues and result in unfair comparisons, we take the same settings as Inter_RAT and FR.

### A.5 MORE RESULTS ABOUT THE ATTACK SUCCESS RATE

More results of the attack success rate are shown in Figure 7.

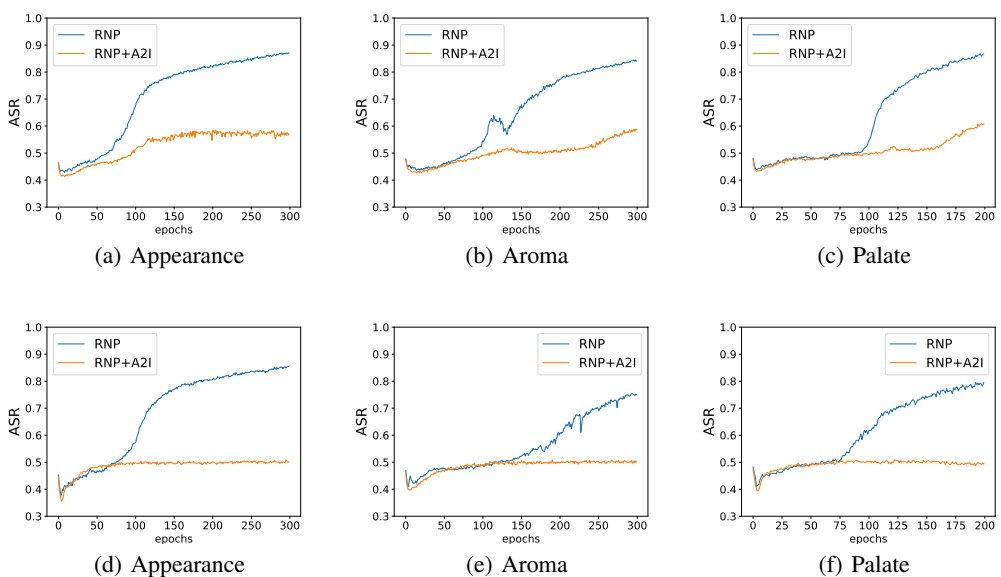

Figure 7: Attack success rate (ASR) on the three aspects of *BeerAdvocate* dataset. The rationale sparsity is about $10\%$ (a,b,c) and $30\%$ (d,e,f).

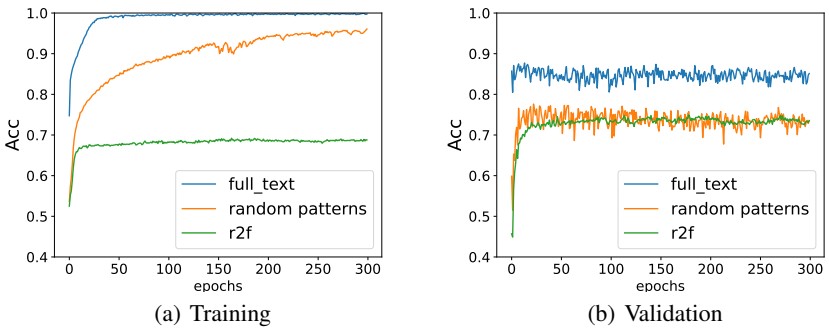

Figure 8: Experiments on the *Appearance* of the BeerAdvocate dataset. The settings are the same as those in Figure 5.

### A.6 MORE RESULTS CORRESPONDING TO FIGURE 5

Figure 5 has shown the results of one aspect of the *BeerAdvocate* dataset. We show the results of the other two aspects in Figure 8 and 9. The green lines can somewhat reflect how much the true sentiment is contained in the randomly selected rationales. And we see that only the true sentiment can generalize to the validation set.

### A.7 EXAMPLES OF SELECTED RATIONALES

We provide two cherry-picked examples of the sampling bias in Figure 10. The sparsity is about $10\%$ (note that $10\%$ is the **average** sparsity across the dataset, and it is manually determined by $s$ in Equation 4 rather than the model's power. So chances are that some texts may have low sparsity and others have high sparsity.).

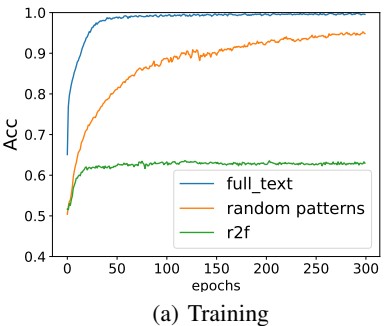 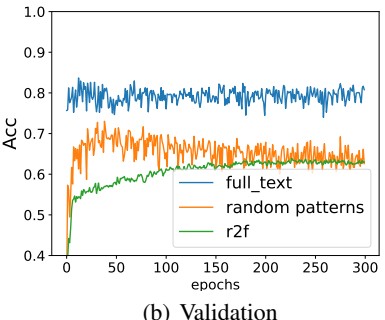

|                  |                  |
| :--------------: | :--------------: |
| (a) Training     | (b) Validation   |

Figure 9: Experiments on the *Palate* of the BeerAdvocate dataset. The settings are the same as those in Figure 5.

---

**Label** (about the beer's appearance): Positive. **Prediction**: Positive.
**Input:** a - murky , semi-opaque honey . low head . s -earthy. plantains , pineapple rind , apricot t - earthy hay and pepper . touch or orange . cilantro . honey . very saison-like . m - medium body . nice carbonation . balanced semi-dry finish . o - nice flavor profile .

(a) RNP

**Label** (about the beer's appearance): Positive. **Prediction**: Positive.
**Input:** a - murky , semi-opaque honey . low head . s -earthy . plantains , pineapple rind , apricot t - earthy hay and pepper . touch or orange . cilantro . honey . very saison-like . m - medium body . nice carbonation . balanced semi-dry finish . o - nice flavor profile .

(b) RNP+A2I

**Label** (about the beer's appearance): Positive. **Prediction**: Positive.
**Input:** bomber poured into my duvel tulip , not sure of the vintage as the bottle was undated , the name of the beer was stenciled on the label rather than hand written ( maybe that helps ) . appearance : amber in color , little to no head even after a hard-pour, what little head there was quickly recedes to a thin ring . smell : sweetness on the nose , maybe some spice ; however , it is the honey like sweetness that is dominant . taste : again , this is a sweet beer and unlike most tripels it remains sweet with very little bitterness on the finish . this has a honey-like flavor mostly . the hops manage to introduce enough balance to keep the beer from becoming cloying . this is an enjoyable beer ( especially if you like a sweeter beer ) but it really would be an odd fit for the tripel style . you do get hints of spice ( and maybe some citrus notes ) as you continue drinking mouthfeel : there is enough carbonation to keep the beer from becoming syrupy and that adds to the overall enjoyment . overall : different for a triple , not bad , but not great . an above average beer for sure . id definitely try it again if they had it on tap when i was in for a visit . abv is well masked .

(c) RNP

**Label** (about the beer's appearance): Positive. **Prediction**: Positive.
**Input:** bomber poured into my duvel tulip , not sure of the vintage as the bottle was undated , the name of the beer was stenciled on the label rather than hand written ( maybe that helps ) . appearance : amber in color , little to no head even after a hard-pour, what little head there was quickly recedes to a thin ring . smell : sweetness on the nose , maybe some spice ; however , it is the honey like sweetness that is dominant . taste : again , this is a sweet beer and unlike most tripels it remains sweet with very little bitterness on the finish . this has a honey-like flavor mostly . the hops manage to introduce enough balance to keep the beer from becoming cloying . this is an enjoyable beer ( especially if you like a sweeter beer ) but it really would be an odd fit for the tripel style . you do get hints of spice ( and maybe some citrus notes ) as you continue drinking mouthfeel : there is enough carbonation to keep the beer from becoming syrupy and that adds to the overall enjoyment . overall : different for a triple , not bad , but not great . an above average beer for sure . id definitely try it again if they had it on tap when i was in for a visit . abv is well masked .

(d) RNP+A2I

Figure 10: Two cherry-picked examples of the sampling bias. The dataset is *Beer-Appearance*. The human-annotated rationales are underlined. The rationales selected by RNP and RNP+A2I are highlighted in red and blue, respectively. The sparsity (average across the dataset) is about $10\%$ (i.e., corresponding to the first row in Table 1).

## B PROOFS

### B.1 THE RELATION BETWEEN ENTROPY AND CROSS-ENTROPY

It is a basic idea in information theory that the entropy of a distribution $P$ is upper bounded by the cross entropy of using $Q$ to approximate it. For any two distribution $P$ and $Q$, we have

$$H_c(P, Q) = H(P) + D_{KL}(P\|Q) \geq H(P), \tag{19}$$

where the subscript $c$ in $H_c(P, Q)$ stands for cross-entropy.

We know that we get the minimum cross entropy when $Q$ is the same as $P$, i.e., $D_{KL}(P\|Q) = 0$. Which means

$$\min H_c(P, Q) = H(P). \tag{20}$$

### B.2 DERIVATION OF EQUATION (8)

To begin with, we need to introduce two fundamental properties from probability theory.

The first property is a general property for conditional probability. If $0 < P(Y = 1) < 1$, then for $\forall p$, if $0 < p < 1$, we can always find a variable $c$, such that $P(Y = 1|c) = p$.

Considering our rationalization situation, we can get the following corollary:

**Corollary 1** *If we can construct $G$ in an arbitrary way, and $0 < P(Y = 1|Z = t) < 1$, then we have*

$$\forall 0 < p < 1, \ \exists g_a \in G, \ P(Y = 1|Z = t, g_a) = p. \tag{21}$$

The second property is also a general property for conditional probability. If $P(Y = 1) = 0$, then for any variable $c$, we always have $P(Y = 1|c) = 0$. This is also a fundamental property in probability theory.

Considering the rationalization situation, let $Z = s_+$, we have

**Corollary 2** *If we can construct $G$ in an arbitrary way, and $P(Y = 0|Z = s_+) = 0$, then we have that there is no $g_a$ that can make $P(Y = 1|Z = t, g_a) > 0$.*

### B.3 More discussion about the singular sentiment assumption

To begin with, we have to acknowledge that this assumption is an idealized scenario. Here are some cases that may break this assumption. Nevertheless, our analysis based on it remains meaningful.

First, the sentiment may be multi-aspect. For example, a person may have positive sentiment about the beer's appearance, while negative sentiment about the taste. If we are discussing the beer's appearance, the review will still be annotated as positive. In such a scenario, the attacker will try to find the negative comment about the taste, and force the predictor to classify it as neutral. However, this is just what we want. It helps the predictor focus not only on the vanilla sentiment, but also on the aspect (which is included in the context of the sentiment) in which we are interested. Since the predictor classifies the comment about the taste as neutral, it will give the only the feedback about the beer's appearance, which can help the generator focus more on the appearance.

Second, the $X$ labelled with $Y = 1$ may be a combination of strong positive sentiment and weak negative sentiment. A dataset may consists of two kind of sentiment: strong and weak, each of which can be divided to positive and negative. The label of $X$ is decided by the strong sentiment. In this scenario, the attacker may find the weak negative sentiment from $X$ labelled with $Y = 1$, and ask the predictor to classify the weak negative sentiment as neutral. There is no denying that weak sentiment and strong sentiment have different styles. Similar to the illustration on multi-aspect sentiment, the attacker here also helps the predictor to focus on strong sentiment and ignore the weak sentiment. As a result, the generator will only select the strong sentiment.

