# OpenReview forum: "Attacking for Inspection and Instruction: Debiasing Self-explaining Text Classification"
_ICLR.cc/2024/Conference — ICLR 2024 Conference Withdrawn Submission_

### Official Review · Reviewer_LkPz · 2023-11-01

**Soundness:** 4 excellent
**Presentation:** 4 excellent
**Contribution:** 2 fair
**Rating:** 6
**Confidence:** 3

**Summary:**

This paper studies the sampling bias problem that arises from the Rationalizing Neural Prediction (RNP) framework for text classifications. The authors demonstrate how sampling bias could be introduced by the explanation generator and how it leads to a bad impact on the label predictor. This paper then proposes to introduce an attacker to inspect the bias and instruct the predictor to prevent the predictor from adopting the bias.

**Strengths:**

Originality: The sampling bias problem of the RNP framework for the text classification setting is first proposed in this paper. Introducing an attacker to alleviate this bias is also novel and interesting.
Quality: Most of the statements are made with sufficient mathematical derivations. In addition, the experiment results provide a practical validation of the statements.
Clarity: The paper has been well written and organized.
Significance: This paper may have a limited impact on the NLP community. The main reason is that this paper focuses on improving the RNP framework for developing self-explainable text classification systems. Although developing such systems is a hot topic in NLP, the RNP framework is just one of the solutions. Also, the RNP framework cannot be well generalized to large pre-trained models such as BERT/GPT (according to the authors), which are more practical and common in recent academic research and industry products.

**Weaknesses:**

The potential of this work is significantly suffered from the fact that the RNP framework cannot be practically aligned with large pre-trained models. In addition, if the authors manually label some data from broader topics and more diverse targets and conduct experiments on them, there would be evidence that the selecting bias is common and inherent exists in the RNP framework, and the proposed method could well alleviate it.

**Questions:**

1.	By providing more high-quality rationales, the predictions should be more accurate. However, I found that sometimes, the baseline methods could better identify rationales with A2I, while the accuracy of predicting labels becomes worse. For example, Beer-Appearance-last grouped row, the F1 score of rationales improves from 72.3 to 74.6, but accuracy drops from 90.9 to 89.7. Similar trends could also be observed from Aroma aspect, for the sixth grouped row, where F1 score improves from 68.4 to 71.2, but accuracy drops from 90.5 to 89.7. Also, in Hotel-Cleanliness, F1 improves from 38.7 to 39.4, while accuracy drops from 96.0 to 95.5. Could the authors provide some insights into this phenomenon?

---

> ### Author Response · Authors · 2023-11-18
> **BERT Encoder Experiments and Clarifications**
>
> Thank you very much for your valuable comments and suggestions!
>
> **Q1.** The potential of this work is significantly suffered from the fact that the RNP framework cannot be practically aligned with large pre-trained models.
>
> **A1.** We agree with you. We have put this issue in the section of limitations, and we leave formally investigating the obstacles in utilizing powerful pre-trained language models under the rationalization framework as the future work.
>
> We have now re-run RNP and FR with the Bert encoder on the three aspects of BeerAdvocate. Although fine-tuning Bert is still very difficult, we find that using a smaller learning rate for the Bert encoder than for the linear layer works somewhat. So we use a learning rate of $1e-5$ for the Bert encoder and $1e-4$ for the linear layer. The maximum sequence length is 256 (this is enough for the Beer dataset). The batch size is 24.
>
> From the second table, we see that Bert\_RNP performs much worse than GRU\_RNP, so there must be some unknown issues prevent RNP to work well with Bert. The sampling bias problem studied in our paper may not be the only obstacle that prevents Bert_RNP to work well, so in most cases, A2I does not improve Bert_RNP a lot.
>
> But from the first table, we see that Bert_FR improves Bert_RNP a lot. So the obstacles introduced by the Bert encoder may have somewhat been overcomed by FR. Under this case, our A2I improves Bert_FR a lot (more than $5\%$ in 6 of 9 settings). We believe that future researchers will further explore harnessing the power of Bert within the RNP framework. At that point, we can also build upon subsequent methods to incorporate our A2I.
>
> Notes: We highlight the results only when FR+A2I gives an improvement over $5\%$ compared to FR
>
> | Bert\_FR |  | Appearance |  |  |  |  |  | Aroma |  |  |  |  |  | Palate |  |  |  |  |
> |---:|:---:|:---:|---:|---:|---:|---:|---:|:---:|---:|---:|---:|---:|---:|:---:|---:|---:|---:|---:|
> |  |  | s | acc | p | r | f1 |  | s | acc | p | r | f1 |  | s | acc | p | r | f1 |
> | $S\approx 10\%$ | FR | 9.9 | 80.2 | 72.5 | 39.0 | 50.8 |  | 11.7 | 80.6 | 56.7 | 41.7 | 48.1 |  | 10.0 | 81.6 | 27.4 | 21.8 | 24.3 |
> |  | FR+A2I | 10.0 | 85.0 | 91.5 | 49.8 | **64.5** |  | 10.2 | 82.3 | 75.9 | 49.0 | **59.5** |  | 9.1 | 83.7 | 35.1 | 25.3 | **29.4** |
> |  |  |  |  |  |  |  |  |  |  |  |  |  |  |  |  |  |  |  |
> | $S\approx 20\%$ | FR | 19.6 | 84.8 | 56.6 | 60.5 | 58.5|  | 19.5 | 83.2 | 43.6 | 53.5 | 48.1 |  | 19.4 | 84.4 | 39.2 | 60.3 | 47.5 |
> |  | FR+A2I | 17.2 | 85.7 | 72.7 | 68.2 | **70.3** |  | 20.2 | 88.7 | 57.6 | 73.3 | **64.5** |  | 20.0 | 85.8 | 42.3 | 66.9 | 51.9 |
> |  |  |  |  |  |  |  |  |  |  |  |  |  |  |  |  |  |  |  |
> | $S\approx 30\%$ | FR | 29.9 | 86.1 | 51.5 | 84.0 | 63.9 |  | 28.7 | 81.6 | 18.1 | 32.7 | 23.3 |  | 28.6 | 82.6 | 12.4 | 28.0 | 17.2 |
> |  | FR+A2I | 30.5 | 87.0 | 51.9 | 86.4 | 64.8 |  | 30.2 | 84.8 | 39.7 | 75.7 | **52.1** |  | 29.0 | 83.2 | 13.1 | 30.0 | 18.2 |
>
> | Bert\_RNP |  | Appearance |  |  |  |  |  | Aroma |  |  |  |  |  | Palate |  |  |  |  |
> |---:|:---:|:---:|---:|---:|---:|---:|---:|:---:|---:|---:|---:|---:|---:|:---:|---:|---:|---:|---:|
> |  |  | s | acc | p | r | f1 |  | s | acc | p | r | f1 |  | s | acc | p | r | f1 |
> | $S\approx 10\%$ | RNP | 10.6 | 83.2 | 38.3 | 22.1 | 28.0 |  | 9.8 | 62.6 | 14.7 | 9.0 | 11.2 |  | 10.0 | 66.0 | 9.8 | 7.7 | 8.6 |
> |  | RNP+A2I | 10.2 | 85.4 | 46.8 | 26.0 | 33.5 |  | 10.4 | 77.5 | 15.0 | 9.8 | 11.9 |  | 10.5 | 76.2 | 10.1 | 8.4 | 9.2 |
> |  |  |  |  |  |  |  |  |  |  |  |  |  |  |  |  |  |  |  |
> | $S\approx 20\%$| RNP | 19.6 | 82.4 | 50.7 | 54.4 | 52.5 |  | 19.3 | 66.5 | 14.7 | 17.8 | 16.1 |  | 20.3 | 70.4 | 10.6 | 17.1 | 13.1 |
> |  | RNP+A2I | 19.8 | 84.5 | 55.6 | 60.2 | 57.8 |  | 19.3 | 78.1 | 15.7 | 19.0 | 17.2 |  | 19.1 | 75.4 | 10.8 | 16.3 | 13.0 |
> |  |  |  |  |  |  |  |  |  |  |  |  |  |  |  |  |  |  |  |
> |$S\approx 30\%$ | RNP | 29.2 | 82.2 | 23.2 | 37.0 | 28.5 |  | 29.6 | 76.3 | 15.8 | 29.5 | 20.6 |  | 29.4 | 72.1 | 11.1 | 25.8 | 15.5 |
> |  | RNP+A2I | 29.9 | 91.1 | 50.7 | 82.7 | 62.9 |  | 29.7 | 77.4 | 19.5 | 36.5 | 25.4 |  | 30.9 | 79.9 | 11.7 | 28.7 | 16.7 |

---

> ### Author Response · Authors · 2023-11-18
> **Other tasks and the prediction accuracy.**
>
> **Q2.** If the authors manually label some data from broader topics and more diverse targets and conduct experiments on them, there would be evidence that the selecting bias is common and inherent exists in the RNP framework, and the proposed method could well alleviate it.
>
> **A2** Thank you for your valuable suggestion. We have now added an experiment conducted with GNNs. We use a very widely used graph classification dataset in the field of explainable GNNs: BA2Motifs. There are labels for the gold rationales: the house motif for class 0 and the cycle motif for class 1. We report the overlap (F1 score) of the selected nodes and gold rationales. The base model for each player is a 2-layer GCN. We report the overlap (F1 score) of the selected nodes and gold rationales. The results are as follows:
>
> | BA2Motifs | S | Acc | P | R | F1 |
> |:---:|:---:|:---:|:---:|:---:|---|
> | RNP | 20.3(2.5) | 95.2(1.9) | 36.5(5.5) | 36.5(2.2) | 36.4(3.8) |
> | RNP+A2I | 20.5(2.3) | 95.2(1.5) | 39.7(3.5) | 40.5(2.9) | **40.0(2.5)** |
> |  |  |  |  |  |  |
> | FR | 20.5(2.3) | 96.4(1.8) | 39.3(5.9) | 40.0(4.9) | 39.6(5.2) |
> | FR+A2I | 20.2(1.5) | 96.5(1.4) | 42.1(2.8) | 42.5(4.0) | **42.3(3.0)** |
>
> Notes: The numbers in "()" are the standard deviations.
>
>
> **Q3.** By providing more high-quality rationales, the predictions should be more accurate.
>
> **A3.** In general, high-quality rationales do improve classification accuracy. However, the prediction accuracy is also influenced by various other factors, leading to some minor fluctuations.
>
> Also, due to the sampling bias we analyzed in this study, poor rationales do not necessarily lead to poor prediction (empirically supported by the experiments in Fig.5). Some experimental results in recent papers also show that better rationale quality does not lead to better prediction. In the paper of CR [A], the best rationale quality is achieved by CR, but the best prediction performance is achieved by FR. In the paper of A2R [B], A2R achieves the best rationale quality, but not the best classification performance.
>
> We also find an interesting paper [C] that argues that spurious features can sometimes improve the accuracy.
>
>
> References
> [A] Towards trustworthy explanation: On causal rationalization. ICML 2023.
> [B] Understanding Interlocking Dynamics of Cooperative Rationalization. NeurIPS 2021.
> [C] Spuriosity Didn't Kill the Classifier: Using Invariant Predictions to Harness Spurious Features. arXiv:2307.

---

### Official Review · Reviewer_H4hs · 2023-11-06

**Soundness:** 2 fair
**Presentation:** 3 good
**Contribution:** 2 fair
**Rating:** 5
**Confidence:** 3

**Summary:**

In this paper, the authors propose a method to address the problem of rationalizing neural prediction. The goal is to use a generator to identify confounding tokens within a language classification task.

Specifically, the authors focus on a binary classification task that assigns a label to a sequence of word tokens. They acknowledge that among these features, only some are causal variables, while others are spurious. The solution is to train a generator that produces a mask to exclude spurious variables. In their work, the authors introduce an adversarial module that learns to select tokens such that they cause a trained predictor to reverse its label, thereby rendering the tokens invariant to the label.

The authors have compared their method with existing studies in the same domain and demonstrated its effectiveness.

**Strengths:**

- Overall this paper is well written, and the author delivered their method pretty clearly.

- The motivation of providing explainable instruction for reasonable about natural language is good, especially for the current era of large language models.

**Weaknesses:**

One of the concerns regarding this work is its contribution; the proposed method seems to be merely an add-on to a specific type of problem. The conclusions drawn from binary classification may not easily generalize to a multi-class setting. In binary classification, selecting the reversed label represents a clear worst-case scenario. However, it is not clear how this approach could extend to multi-class cases. In the appendix, the authors provide formulations for multi-class scenarios in equations (17) and (18). I encourage the authors to deliberate on the specific method for "choosing the $Y' \neq Y$—should this $Y$
  be sampled from a uniform distribution, for instance? Moreover, the authors' method assumes the availability of a balanced dataset. How would the algorithm be modified in the presence of imbalanced labels?

**Questions:**

Reasoning from language could be complicated, there could be more structured knowledge in one sentence beyond confounding information. Sometimes the same words may imply opposite meanings under different contexts. I am wondering how the authors are going to address these more complicated problems in NLP.

**Details Of Ethics Concerns:**

No issue

---

> ### Author Response · Authors · 2023-11-18
> **Thank you very much for the thoughtful review!**
>
> We are really grateful for your constructive feedback and guidance!
>
> **Q1.** One of the concerns regarding this work is its contribution; the proposed method seems to be merely an add-on to a specific type of problem.
>
> **A1.** We are sorry for the confusion, but we think this is a misunderstanding. Aside from the proposed method, the identification of the specific sampling bias problem is also one of our contributions. We are the first one to identify this problem, and this finding is supported by theoretical analysis (Section 4.2) and empirical evidence (Section 5, Fig.5, Fig.6, Fig.7-9).
>
> Although the proposed method is an add-on to RNP and its variants, but on the other hand, our method is flexible enough to benefit a number of approaches in this field.
>
> **Q2.** The conclusions drawn from binary classification may not easily generalize to a multi-class setting. In binary classification, selecting the reversed label represents a clear worst-case scenario. However, it is not clear how this approach could extend to multi-class cases. In the appendix, the authors provide formulations for multi-class scenarios in equations (17) and (18). I encourage the authors to deliberate on the specific method for "choosing the $Y'\neq Y$ —should this $Y$ be sampled from a uniform distribution, for instance?
>
> **A2.** Thank you for your valuable suggestion. We would like to first clarify a misunderstanding, the $Y$ in Equation 17 is the true label of $X$. $Y'\neq Y$ means the prediction does not match the true label. The $\min_{Y'}$ operator means we want to find a $Y'$ that makes the attack easiest.
>
> Here is the detailed clarification on Equation 17:
> The general idea is the same as for binary classification:  Consider an n-class classification task. We denote $X^j$ as the text of the $j$-th class. For the $i$-th class, the goal of the attacker  is to find $Z_A$ from those $X^j$ ($j$ can be any value not equal to $i$, i.e., $X^j$ doesn't belong to the $i$-th class) and get the predictor to predict $Z_A$ as the $i$-th class. Equation 17 is a practical implementation of this idea: for an aribitrary data point $X^Y$, the attacker finds the attack rationale $Z_A=f_a(X^Y)$, and the predictor's output is $f_p(Z_A)$. The significance of the $\min_{\theta_a}$ operator in Equation 17 is such that $f_p(Z_A)$ is categorized as different from $Y$. And the $\min_{Y'}$ operator is to see which class $f_p(Z_A)$ is closest to, making the attack easier. If $f_p(Z_A)$ is successfully classified as class $Y'$, then we think the predictor uses some trivial patterns that do not belong exclusively to the $Y'$-th class to make predictions about $Y'$.
> You can understand it in a simpler way: $Y,Y'$ in Equation 17 correspond to the above $j$ adn $i$, respectively.  There is a simple way to bridge binary and multi-class classification: In binary classification, $[i,j]=[1,0]$ or $[0,1]$. In n-class calssification, $i,j$ can be any values as long as $i\neq j, 1\leq i,j\leq n$.
> And Equation 18 (the instruction term) is no different from the second term of Equation 5.
>
> Also, **our analysis about the problem is not limited to binary classification tasks.** The most important analysis about the root of the problem is Equation 7. Here we consider an n-class classification task.
> **Equation 7 can be extended as follows:**
> The setup: $T\in \{t_1,t_2,...,t_m\}$ ($m$ can be an arbitrary integer). $\forall i \in [1,m], \ P(Y)=P(Y|t_i)$, which means $T$ is a trivial pattern independent with $Y$.
> The first row of Equation 7 becomes: $\exist 1\leq j\leq n,\  1\leq i\leq m,\  P(Y=j|Z=t_i,g)>P(Y=j),$ which means that $t_i$ becomes an indicator for the $j$-th class under the condition of the generator.
>
>
> **Q3.**  Moreover, the authors' method assumes the availability of a balanced dataset. How would the algorithm be modified in the presence of imbalanced labels?
>
> **A3.** I guess your question stems from our assumption $P(Y=1)=P(Y=0)=0.5$ in Equation 7. This assumption is made for the sake of theoretical analysis simplicity. For example, if $P(Y=1)>P(Y=0)$ in the original dataset, then a predictor does not need any input to get a accuracy higher than $50\%$. For unbalanced datasets, we can employ existing methods designed to handle unbalanced labels (e.g., random under-sampling). In fact, the BeerAdvocate dataset we used is a very unbalanced dataset, and the method we used to handle it is random under-sampling. Here are the statistics for the training data of BeerAdvocate:
>
> | aspect | positive | negative |
> |:---:|---|---|
> | appearance | 53114 | 16891 |
> | aroma | 46386 | 15169 |
> | palate | 47592 | 13652 |

---

### Official Review · Reviewer_QZhC · 2023-11-06

**Soundness:** 3 good
**Presentation:** 3 good
**Contribution:** 3 good
**Rating:** 6
**Confidence:** 3

**Summary:**

The paper addresses the task of eXplainable Artificial Intelligence (XAI) where the goal is to increase the transparency of deep learning models to enhance trust in their decisions regarding fairness, security, and robustness. It explores a self-explaining framework called Rationalizing Neural Predictions (RNP) used in NLP models, which employs a cooperative game involving a generator and a predictor.

It identifies a potential sampling bias issue in the RNP framework, where the generator might select semantically unrelated trivial patterns as explanations, leading to implausible explanations. The paper proposes an adversarial game-based approach to inspect and identify this bias, and introduces a method to instruct the game to debias the predictor by penalizing it when it learns from the bias.

Experimental results demonstrate the existence of sampling bias and the effectiveness of the inspection and instruction methods, which are model-agnostic and improve the performance of self-explaining models.

**Strengths:**

- Originality: the paper proposes an interesting strategy to identify the bias problem within self-explanation, and introduces an efficient combination of an adversarial approach, and the development of an instruction objective for mitigating bias.

- Quality: the paper runs a decent set of experiments to evaluate their method against existing ones.

- Clarity: the presentation of the methodology, the pipeline and the experimental results are well-structured and easy to follow.

- Significance: contributions made to self-explaining rationalization and interpretable machine learning have wild impact in the literature, and the fact the authors achieved good results with their method is significant

**Weaknesses:**

- No code to verify the results

-  While the paper discusses the theoretical aspects of sampling bias and introduces a solution for self-explanation, it falls short in discussing the real-world implications of this bias in AI applications. Providing concrete examples or case studies of how sampling bias can impact decision-making systems would enhance the paper's practical relevance.

- The datasets BeerAdvocate and HotelReview seem small and basic. I am curious to see how this method performs on larger scale datasets like CIFAR10

- No reporting of the mean and standard deviation of multiple experiments to see if the results are significant

**Questions:**

Please address the weaknesses above.

---

> ### Author Response · Authors · 2023-11-18
> **Thank you very much for the insightful review and suggestions!**
>
> We really appreciate your valuable comments and suggestions!
>
> **Q1.** No code to verify the results
>
> **A1.** We have now updated the code and the instructions at the general response. They are only visible to reviewers.
>
> **Q2.** While the paper discusses the theoretical aspects of sampling bias and introduces a solution for self-explanation, it falls short in discussing the real-world implications of this bias in AI applications. Providing concrete examples or case studies of how sampling bias can impact decision-making systems would enhance the paper's practical relevance.
>
> **A2.** Thank you for your suggestion. We have now added two case studies in Fig.10 of Appendix A.7.
>
> **Q3.** The datasets BeerAdvocate and HotelReview seem small and basic. I am curious to see how this method performs on larger scale datasets like CIFAR10.
>
> **A3.** While BeerAdvocate is relatively small (about 30,000 texts for each aspect), the HotelReview is much larger. Hotel-Service has more than 100,000 texts and Hotel-Cleanliness has more than 150,000 texts. We'd like to conduct experiments on CIFAR10, but the problem is that it does not have human-annotated rationales for evaluation. As we have mentioned in Section 5.2.1, the sampling bias problem makes the prediction performance not a good metric.
>
> As an alternative, we now add experiments on a graph neural network dataset to validate the generalization capabilities of the method.  We use a wiedly used dataset in the field of explainable GNNs: BA2Motifs. This is a graph classification dataset and has labeled gold rationales for evaluation. The base model for each player is a 2-layer GCN. We report the overlap (F1 score) of the selected nodes and gold rationales.
> The results are as follows:
>
> | BA2Motifs | S | Acc | P | R| F1 |
> |:---:|:---:|:---:|:---:|:---:|---|
> | RNP | 20.3(2.5) | 95.2(1.9) | 36.5(5.5) | 36.5(2.2) | 36.4(3.8) |
> | RNP+A2I | 20.5(2.3) | 95.2(1.5) | 39.7(3.5) | 40.5(2.9) | **40.0(2.5)** |
> |  |  |  |  |  |  |
> | FR | 20.5(2.3) | 96.4(1.8) | 39.3(5.9) | 40.0(4.9) | 39.6(5.2) |
> | FR+A2I | 20.2(1.5) | 96.5(1.4) | 42.1(2.8) | 42.5(4.0) | **42.3(3.0)** |
>
> Notes: The numbers in "()" are the standard deviations.
>
> **Q4.** No reporting of the mean and standard deviation of multiple experiments to see if the results are significant
>
> **A4.** We now report the standard deviation. In our original experiments, we use a fixed random seed of 12252018 (inherited from the code provided by our baseline FR), because we think that experiments with 12 different settings (beer: 3 aspects $*$ 3 sparsity, hotel: 3 aspects) under the same random seed are somewhat sufficient to verify the stability of the models. Here we rerun RNP and RNP+A2I with 4 additional seeds and report the standard deviation over the five random seeds. Notes: The format of the numbers is "avg(std)".
>
> |  |  | Appearance |  |  |  |  |  | Aroma |  |  |  |  |  | Palate |  |  |  |  |
> |---:|:---:|:---:|---:|---:|---:|---:|---:|:---:|---:|---:|---:|---:|---:|:---:|---:|---:|---:|---:|
> |  |  | s | acc | p | r | f1 |  | s | acc | p | r | f1 |  | s | acc | p | r | f1 |
> | $S\approx 10\%$  | RNP | 9.0(1.2) | 81.5(1.5) | 83.4(8.0) | 40.3(5.5) | 54.2(5.9) |  | 9.2(1.2) | 83.7(1.7) | 84.1(2.7) | 49.7(5.5) | 62.3(4.1) |  | 9.7(0.3) | 83.2(1.8) | 69.1(2.1) | 53.8(1.9) | 60.5(1.8) |
> |  | RNP+A2I | 10.0(0.6) | 82.6(1.3) | 82.2(3.6) | 44.8(1.0) | 58.1(0.7) |  | 9.9(0.3) | 83.8(2.0) | 84.5(1.0) | 53.9(1.2) | 65.8(0.9) |  | 10.1(0.5) | 85.4(1.1) | 69.3(2.3) | 56.3(1.6) | 62.1(1.3) |
> |  |  |  |  |  |  |  |  |  |  |  |  |  |  |  |  |  |  |  |
> | $S\approx 20\%$  | RNP | 19.5(0.3) | 83.3(1.4) | 69.2(2.6) | 73.1(3.7) | 71.1(3.1) |  | 21.0(0.7) | 85.8(1.2) | 43.9(2.7) | 59.2(2.0) | 50.4(2.5) |  | 19.2(0.8) | 85.3(1.9) | 47.0(2.0) | 72.7(5.0) | 57.0(2.9) |
> |  | RNP+A2I | 20.0(0.1) | 85.2(2.6) | 72.6(0.9) | 78.6(1.0) | 75.5(1.0) |  | 19.5(0.3) | 86.3(1.7) | 50.1(1.0) | 62.7(1.4) | 55.6(1.1) |  | 18.8(0.8) | 86.2(0.6) | 48.4(3.0) | 72.8(2.9) | 58.2(2.9) |
> |  |  |  |  |  |  |  |  |  |  |  |  |  |  |  |  |  |  |  |
> | $S\approx 30\%$  | RNP | 30.5(1.1) | 85.5(2.4) | 55.9(2.6) | 92.2(3.2) | 69.6(2.7) |  | 31.2(0.4) | 85.9(3.7) | 39.2(1.8) | 78.5(3.1) | 52.3(2.3) |  | 29.0(0.9) | 86.1(2.6) | 31.1(2.7) | 72.6(6.1) | 43.6(3.7) |
> |  | RNP+A2I | 29.7(0.2) | 85.7(1.7) | 59.5(0.7) | 95.5(1.3) | 73.4(0.9) |  | 29.4(1.1) | 88.0(1.2) | 44.8(1.6) | 84.6(3.7) | 58.5(2.0) |  | 28.5(0.9) | 86.4(0.8) | 32.5(1.7) | 74.7(5.0) | 45.3(2.5) |
>
>
> Due to the large amount of GPU resources consumed by Bert related experiments, the random seed experiments for FR are still in preparation.

---

### Official Review · Reviewer_MTBk · 2023-11-08

**Soundness:** 2 fair
**Presentation:** 2 fair
**Contribution:** 2 fair
**Rating:** 5
**Confidence:** 3

**Summary:**

The paper presents an interesting approach to addressing the issue of sampling bias in self-explaining text classification models. The authors propose a method called Attack to Inspection and Instruction (A2I), which uses an adversarial game to inspect and correct the predictor's behavior in a Rationalizing Neural Predictions (RNP) framework. The paper is well-motivated, and the problem of sampling bias in self-explaining models is a relevant and important one in the field of explainable AI.

**Strengths:**

1. The paper addresses a significant problem in the field of explainable AI, which is the potential for sampling bias to lead to incorrect correlations between selected explanations and labels.
2. The authors provide a thorough theoretical motivation for their approach, explaining how the adversarial game can detect and mitigate sampling bias.
3. The experiments conducted on two real-world benchmarks demonstrate the effectiveness of the proposed method, with significant improvements in rationale quality over the baseline RNP model and other advanced methods.

**Weaknesses:**

1. The paper primarily focuses on binary classification tasks, and it is not clear how well the proposed method would generalize to multi-class classification or other types of machine learning tasks.
2. While the authors mention that the proposed method is model-agnostic, the experiments are limited to the RNP framework and its variants. It would be beneficial to see the method applied to other types of models to assess its generalizability.
3. The paper could benefit from a more detailed discussion on the limitations of the proposed method, including potential scenarios where the adversarial game might fail to detect certain types of biases or where the instruction phase might not effectively debias the predictor.
4. The use of GRUs and GloVe embeddings, while understandable for comparison purposes, may not reflect the current state-of-the-art in NLP, where transformer-based models like BERT are prevalent. It would be interesting to see how the proposed method performs with such models.

**Questions:**

None

---

> ### Author Response · Authors · 2023-11-18
> **About multi-class classification, other tasks, and limitations**
>
> We greatly appreciate your detailed review and constructive feedback on our paper!
>
> *Q1.** The paper primarily focuses on binary classification tasks, and it is not clear how well the proposed method would generalize to multi-class classification or other types of machine learning tasks.
>
> **A1.** Although we use binary classification datasets in our experiments (since there are no proper multi-class classification datasets that have human-annotated rationales), we have also provided formulations for multi-class scenarios in Appendix A.2.  The general idea is the same as for binary classification:  Consider an n-class classification task. We denote $X^j$ as the text of the $j$-th class. For the $i$-th class, the goal of the attacker  is to find $Z_A$ from those $X^j$ ($j$ can be any value not equal to $i$, i.e., $X^j$ doesn't belong to the $i$-th class) and get the predictor to predict $Z_A$ as the $i$-th class. Equation 17 is a practical implementation of this idea: for an aribitrary data point $X^Y$, the attacker finds the attack rationale $Z_A=f_a(X^Y)$, and the predictor's output is $f_p(Z_A)$. The significance of the $\min_{\theta_a}$ operator in Equation 17 is such that $f_p(Z_A)$ is categorized as different from $Y$. And the $\min_{Y'}$ operator is to see which class $f_p(Z_A)$ is closest to, making the attack easier. If $f_p(Z_A)$ is successfully classified as class $Y'$, then we think the predictor uses some trivial patterns that do not belong exclusively to the $Y'$-th class to make predictions about $Y'$.
> You can understand it in a simpler way: $Y,Y'$ in Equation 17 correspond to the above $j$ adn $i$, respectively.  There is a simple way to bridge binary and multi-class classification: In binary classification, $[i,j]=[1,0]$ or $[0,1]$. In n-class calssification, $i,j$ can be any values as long as $i\neq j, 1\leq i,j\leq n$.
> And Equation 18 (the instruction term) is no different from the second term of Equation 5.
>
> Also, **our analysis about the problem is not limited to binary classification tasks.** The most important analysis about the root of the problem is Equation 7. Here we consider an n-class classification task.
> **Equation 7 can be extended as follows:**
> The setup: $T\in \{t_1,t_2,...,t_m\}$ ($m$ can be an arbitrary integer). $\forall i \in [1,m], \ P(Y)=P(Y|t_i)$, which means $T$ is a trivial pattern independent with $Y$.
> The first row of Equation 7 becomes: $\exist 1\leq j\leq n,\  1\leq i\leq m,\  P(Y=j|Z=t_i,g)>P(Y=j),$ which means that $t_i$ becomes an indicator for the $j$-th class under the condition of the generator.
>
> **Experiments on GNNs:**
> We have now generalized our method to other machine learning tasks. In particular, we extend it to explainable graph neural networks. We conduct experiments on BA2Motifs, which is a widely used dataset in the field of explainable GNNs. It's a graph classification dataset. There are labels for the gold rationales: the house motif for class 0 and the cycle motif for class 1. The base model for each player is a 2-layer GCN. We report the overlap (F1 score) of the selected nodes and gold rationales.
>
> |  | s | acc | p | r | f1 |
> |:---:|:---:|:---:|:---:|:---:|---|
> | RNP | 20.3(2.5) | 95.2(1.9) | 36.5(5.5) | 36.5(2.2) | 36.4(3.8) |
> | RNP+A2I | 20.5(2.3) | 95.2(1.5) | 39.7(3.5) | 40.5(2.9) | **40.0(2.5)** |
> |  |  |  |  |  |  |
> | FR | 20.5(2.3) | 96.4(1.8) | 39.3(5.9) | 40.0(4.9) | 39.6(5.2) |
> | FR+A2I | 20.2(1.5) | 96.5(1.4) | 42.1(2.8) | 42.5(4.0) | **42.3(3.0)** |
>
> Notes: The numbers in "()" are the standard deviations.
>
> **Q2.** While the authors mention that the proposed method is model-agnostic, the experiments are limited to the RNP framework and its variants. It would be beneficial to see the method applied to other types of models to assess its generalizability.
>
> **A2.** We are grateful for the suggestion, but we think there is a misunderstanding. RNP is not a niche field, as mentioned in the related work section, with new research in this domain emerging every year. This is attributed to the fact that the cooperative game proposed by RNP is a mainstream approach in NLP for constructing self-explanatory models. When we refer to "model-agnostic," we are referring specifically to RNP and its variants. Now, we have added additional experiments to show that the method can be generalized to GNNs, which is shown in **A1** above.
>
> **Q3.** The paper could benefit from a more detailed discussion on the limitations of the proposed method, including potential scenarios where the adversarial game might fail to detect certain types of biases or where the instruction phase might not effectively debias the predictor.
>
> **A3.** Thanks a lot for your suggestions. Since we are focusing on the bias introduced by the generator's selection, one type of the bias that we cannot deal with is the bias already exist in the dataset. We will add it to the present limitations.

---

> ### Author Response · Authors · 2023-11-18
> **BERT Encoder Experiments and Clarifications**
>
> **Q4.**  The use of GRUs and GloVe embeddings, while understandable for comparison purposes, may not reflect the current state-of-the-art in NLP, where transformer-based models like BERT are prevalent. It would be interesting to see how the proposed method performs with such models.
>
> **A4.** Thanks for your suggestion. We have now re-run RNP and FR with the Bert encoder on the three aspects of BeerAdvocate. Although fine-tuning Bert is still very difficult, we find that using a smaller learning rate for the Bert encoder than for the linear layer works somewhat. So we use a learning rate of $1e-5$ for the Bert encoder and $1e-4$ for the linear layer. The maximum sequence length is 256 (this is enough for the Beer dataset). The batch size is 24.
>
> From the second table, we see that Bert\_RNP performs much worse than GRU\_RNP, so there must be some unknown issues prevent RNP to work well with Bert. The sampling bias problem studied in our paper may not be the only obstacle that prevents Bert_RNP to work well, so in most cases, A2I does not improve Bert_RNP a lot.
>
> But from the first table, we see that Bert_FR improves Bert_RNP a lot. So the obstacles introduced by the Bert encoder may have somewhat been overcomed by FR. Under this case, our A2I improves Bert_FR a lot (more than $5\%$ in 6 of 9 settings). We believe that future researchers will further explore harnessing the power of Bert within the RNP framework. At that point, we can also build upon subsequent methods to incorporate our A2I.
>
> Notes: We highlight the results only when FR+A2I gives an improvement over $5\%$ compared to FR.
>
> | Bert\_FR |  | Appearance |  |  |  |  |  | Aroma |  |  |  |  |  | Palate |  |  |  |  |
> |---:|:---:|:---:|---:|---:|---:|---:|---:|:---:|---:|---:|---:|---:|---:|:---:|---:|---:|---:|---:|
> |  |  | s | acc | p | r | f1 |  | s | acc | p | r | f1 |  | s | acc | p | r | f1 |
> | $S\approx 10\%$ | FR | 9.9 | 80.2 | 72.5 | 39.0 | 50.8 |  | 11.7 | 80.6 | 56.7 | 41.7 | 48.1 |  | 10.0 | 81.6 | 27.4 | 21.8 | 24.3 |
> |  | FR+A2I | 10.0 | 85.0 | 91.5 | 49.8 | **64.5** |  | 10.2 | 82.3 | 75.9 | 49.0 | **59.5** |  | 9.1 | 83.7 | 35.1 | 25.3 | **29.4** |
> |  |  |  |  |  |  |  |  |  |  |  |  |  |  |  |  |  |  |  |
> | $S\approx 20\%$ | FR | 19.6 | 84.8 | 56.6 | 60.5 | 58.5|  | 19.5 | 83.2 | 43.6 | 53.5 | 48.1 |  | 19.4 | 84.4 | 39.2 | 60.3 | 47.5 |
> |  | FR+A2I | 17.2 | 85.7 | 72.7 | 68.2 | **70.3** |  | 20.2 | 88.7 | 57.6 | 73.3 | **64.5** |  | 20.0 | 85.8 | 42.3 | 66.9 | 51.9 |
> |  |  |  |  |  |  |  |  |  |  |  |  |  |  |  |  |  |  |  |
> | $S\approx 30\%$ | FR | 29.9 | 86.1 | 51.5 | 84.0 | 63.9 |  | 28.7 | 81.6 | 18.1 | 32.7 | 23.3 |  | 28.6 | 82.6 | 12.4 | 28.0 | 17.2 |
> |  | FR+A2I | 30.5 | 87.0 | 51.9 | 86.4 | 64.8 |  | 30.2 | 84.8 | 39.7 | 75.7 | **52.1** |  | 29.0 | 83.2 | 13.1 | 30.0 | 18.2 |
>
> | Bert\_RNP |  | Appearance |  |  |  |  |  | Aroma |  |  |  |  |  | Palate |  |  |  |  |
> |---:|:---:|:---:|---:|---:|---:|---:|---:|:---:|---:|---:|---:|---:|---:|:---:|---:|---:|---:|---:|
> |  |  | s | acc | p | r | f1 |  | s | acc | p | r | f1 |  | s | acc | p | r | f1 |
> | $S\approx 10\%$ | RNP | 10.6 | 83.2 | 38.3 | 22.1 | 28.0 |  | 9.8 | 62.6 | 14.7 | 9.0 | 11.2 |  | 10.0 | 66.0 | 9.8 | 7.7 | 8.6 |
> |  | RNP+A2I | 10.2 | 85.4 | 46.8 | 26.0 | 33.5 |  | 10.4 | 77.5 | 15.0 | 9.8 | 11.9 |  | 10.5 | 76.2 | 10.1 | 8.4 | 9.2 |
> |  |  |  |  |  |  |  |  |  |  |  |  |  |  |  |  |  |  |  |
> | $S\approx 20\%$| RNP | 19.6 | 82.4 | 50.7 | 54.4 | 52.5 |  | 19.3 | 66.5 | 14.7 | 17.8 | 16.1 |  | 20.3 | 70.4 | 10.6 | 17.1 | 13.1 |
> |  | RNP+A2I | 19.8 | 84.5 | 55.6 | 60.2 | 57.8 |  | 19.3 | 78.1 | 15.7 | 19.0 | 17.2 |  | 19.1 | 75.4 | 10.8 | 16.3 | 13.0 |
> |  |  |  |  |  |  |  |  |  |  |  |  |  |  |  |  |  |  |  |
> |$S\approx 30\%$ | RNP | 29.2 | 82.2 | 23.2 | 37.0 | 28.5 |  | 29.6 | 76.3 | 15.8 | 29.5 | 20.6 |  | 29.4 | 72.1 | 11.1 | 25.8 | 15.5 |
> |  | RNP+A2I | 29.9 | 91.1 | 50.7 | 82.7 | 62.9 |  | 29.7 | 77.4 | 19.5 | 36.5 | 25.4 |  | 30.9 | 79.9 | 11.7 | 28.7 | 16.7 |

---

### Official Review · Reviewer_TefW · 2023-11-09

**Soundness:** 2 fair
**Presentation:** 1 poor
**Contribution:** 2 fair
**Rating:** 3
**Confidence:** 3

**Summary:**

The authors study the problem of self-explainable models through the lens of the Rationalizing Neural Predictions (RNP) framework, where a generator which selects a subset of the input sequence is trained jointly with the predictor to produce an extractive rationale. Here, they look to explain and tackle one particular problem with RNP -- that it may degenerate into the generator selecting a special semantically unmeaningful token, which the predictor learns to associate with a particular label. The authors first theoretically study the problem through the perspective of sampling bias. Then, they propose a method based on training an attacker which tries to produce a justification for the opposite label, and regularizing the generator with this justification. They show that their method outperforms the baselines on typical RNP datasets.

**Strengths:**

- The authors beat the baselines on typical RNP datasets.
- The proposed method is intuitive at a high level.

**Weaknesses:**

1. The authors do not sufficiently show that this degeneration is an issue empirically in my opinion. To start, the authors should show a few real examples where vanilla RNP gives a nonsense justification while the predictor still outputs the correct label; and show that RNP + A2I fixes these cases. In addition, the authors could consider plotting a histogram of the length of the rationale (for RNP and RNP+A2I), and showing that samples with short justifications correspond to degenerate cases (e.g. the punctuation example). Overall, the sparsity of the A2I augmented models (in Table 1) do not seem significantly different from the sparsity of the base models, and so I am not convinced that A2I solves the issue presented.

2. The proposed method makes sense, but there are several much simpler solutions that the authors should try and compare with. First, it seems to me that the root cause of the problem is that the generator is overpowered -- it's able to internally detect the label, and then feed special tokens correlated with the label to the predictor which do not have semantic meaning. As such, some simple solutions would be to reduce the capacity of the generator; add regularization to the generator; or to train the generator and predictor in an alternating fashion, with more steps for the predictor. The final suggestion is similar to how GANs are trained. In addition, the problem examined is very similar to mode collapse in GANs, and some of the solutions there (e.g. a diversity regularizer [1]) could work as well.


3. There are a few edge cases which I am not convinced that A2I will be able to fix. Primarily, these deal with the circumstance where, in the toy example, the $t_+$ do not appear in the negative samples, and vice versa -- so $t_+$ is a token that appears almost exclusively in positive examples, and $t_-$ is a token that appears almost exclusively in negative examples. Such spurious correlations have been found in natural language tasks [3-4]. In these cases, it seems like the attacker would not be able to choose the corresponding token, and would thus still output random noise.

4. Another concern deals with the singular sentiment assumption. This seems like a strong assumption that is very dataset and task specific, and the authors already discuss its failure modes in the appendices. The presence of negation seems to be another case where the assumption would be violated. As such, I am not convinced in the generalizability of the method to other datasets and tasks.  Regardless, the authors should formulate this assumption mathematically in the text.


5. Overall, the clarity of the paper could be improved. Some of the formulation sections are hard to parse. For example, the authors formulate the problem as one of sampling bias, which makes sense intuitively. However, the mathematical formulation and causal graphs for this section don't follow the prior work in sampling bias [2].


6. The utility of the method is limited in the era of large pre-trained LLMs, which would achieve very high _zero-shot_ accuracy on all of the sentiment tasks evaluated, likely even higher than the GloVe + GRU networks studied in the paper. Such LLMs also have the capability of explaining its own reasoning (as the authors have referenced). To improve the significance of the method, the authors should consider applying their method to finetune a large-scale LLM (though the authors mention that even finetuning BERT is challenging for RNP). They could also consider applying it to images and graphs, as described in the introduction.

7. The authors do not show any confidence intervals for their results, so it is unclear whether performance gains are statistically significant. They also only evaluate on two datasets, though these seem to be standard datasets in the RNP community.


[1] Diversity-Sensitive Conditional Generative Adversarial Networks. ICLR 2019.

[2] Controlling Selection Bias in Causal Inference. AISTATS 2012.

[3] An empirical study on robustness to spurious correlations using pre-trained language models. TACL 2020.

[4] On Feature Learning in the Presence of Spurious Correlations. NeurIPS 2022.

**Questions:**

Please address the weaknesses above.

---

> ### Author Response · Authors · 2023-11-18
> **Concerns about the empirical support for the existence of the problem.**
>
> We sincerely appreciate your efforts to help improve the quality of this study. Here are some more detailed clarifications that may address your concerns.
>
> **Q1.** The authors do not sufficiently show that this degeneration is an issue empirically in my opinion. To start, the authors should show a few real examples where vanilla RNP gives a nonsense justification while the predictor still outputs the correct label; and show that RNP + A2I fixes these cases. In addition, the authors could consider plotting a histogram of the length of the rationale (for RNP and RNP+A2I), and showing that samples with short justifications correspond to degenerate cases (e.g. the punctuation example). Overall, the sparsity of the A2I augmented models (in Table 1) do not seem significantly different from the sparsity of the base models, and so I am not convinced that A2I solves the issue presented.
>
> **A1.1. (The authors do not sufficiently show that this degeneration is an issue empirically in my opinion.)**
> Sorry for the confusion, we indeed did not show it in the introduction. But we have shown it with practical experiments in Section 5.
>
> We first show the risk of this bias with experiments designed from three perspectives in Section 5.1 (i.e., Fig.5, Fig.8, and Fig.9).
> We present three types of prediction accuracies for the BeerAdvocate dataset: (1) A predictor trained
> with the full input text. (2) A predictor trained with randomly selected patterns. For the generator, we
> remove the other objectives and only train it with the sparsity constraints. In other words, the generator
> is trained to randomly select 10% of the input text, and the predictor is then trained to classify using
> these randomly selected texts. (3) We use the randomly selected texts from (2) to feed the predictor
> trained in (1).
> The result for the Aroma aspect is shown in Figure 5. From Figure 5(a), we observe that even with the
> randomly selected patterns (i.e., patterns unlikely to contain real rationales), the predictor can still
> achieve a very high prediction accuracy (represented by the orange line, approximately 95%). This
> accuracy is close to that of the classifier trained with the full texts. A followed question is: Does this
> result suggest that the 10% randomly selected patterns already contain enough sentiment inclination
> for classification? The answer is no. Consider the green line, which represents the outcome when
> we feed the randomly selected texts to the predictor denoted by the blue line. We observe that the
> green line indicates a significantly lower accuracy (about 58%), implying that the randomly selected
> patterns contain only minimal sentiment information. Thus, the orange predictor incorrectly treats
> certain randomly selected trivial patterns as indicative features.
>
> Then, we show the existence of the bias in the normally trained RNP framework with Attack Success Rate (ASR) in Section 5.2 (i.e., Fig.6 and Fig.7). In Fig.6(a), the attack success rate is very high (about $85\%$), which indicates that the predictor indeed utilizes some category-agnostic features for classification, and the attacker has successfully identified these features.
>
> **A1.2 (the authors should show a few real examples where vanilla RNP gives a nonsense justification while the predictor still outputs the correct label; and show that RNP + A2I fixes these cases.)**
> Thank you for your suggestions. We have now added two examples of the sampling bias in Figure 10 of Appendix A.7.  The sparsity is about $10\%$ (note that $10\%$ is the **average** sparsity across the dataset, and it is manually determined by $s$ in Equation 4 rather than the model's power. So it is possible that some texts may have low sparsity and others have high sparsity.).
>
> **A1.3 (In addition, the authors could consider plotting a histogram of the length of the rationale (for RNP and RNP+A2I), and showing that samples with short justifications correspond to degenerate cases (e.g. the punctuation example). Overall, the sparsity of the A2I augmented models (in Table 1) do not seem significantly different from the sparsity of the base models, and so I am not convinced that A2I solves the issue presented.).**
> We are sorry, but we think this is a misunderstanding. The sparsity in Table 1 is the average sparsity over the entir dataset, and it is manually determined by $s$ in Equation 4 rather than the model's power. The punctuation example is just an imaginary toy example to better understand this problem. You can also refer to the example in our Fig.2, where the sentence "I went to a hotel yesterday" is a trivial pattern and plays the same role as "." in the punctuation example. In fact, short justifications do not correspond to degenerate cases and vice versa.

---

> ### Author Response · Authors · 2023-11-18
> **Other methods to consider.**
>
> **Q2.** The proposed method makes sense, but there are several much simpler solutions that the authors should try and compare with. First, it seems to me that the root cause of the problem is that the generator is overpowered -- it's able to internally detect the label, and then feed special tokens correlated with the label to the predictor which do not have semantic meaning. As such, some simple solutions would be to reduce the capacity of the generator; add regularization to the generator; or to train the generator and predictor in an alternating fashion, with more steps for the predictor. The final suggestion is similar to how GANs are trained. In addition, the problem examined is very similar to mode collapse in GANs, and some of the solutions there (e.g. a diversity regularizer [1]) could work as well.
>
> **A2.** Thanks you for your insightful suggestions. Our initial focus was on training a good predictor to guide the generator in producing appropriate rationales. This is because we believe that the generator must first identify the sentiment and then select the rationales that represent the sentiment. So, our focus is on avoiding the second part of your suggestion (feed special tokens correlated with the label to the predictor which do not have semantic meaning) .
> Also, experiments in the reviewer-author discussion of FR (one of our baselines) showed that adding more layers to the encoder of the RNP's generator does not necessarily decrease its performance (more powerful generator does not lead to worse results, so reducing the capacity may not be the first choice for us). Finally, we want to point out that the root cause is not simply that the generator is overpowered. We think the root is $Y\perp Z\nRightarrow Y\perp Z|g$, meaning that any random bias in the generator's selection can make a trivial pattern (independent with $Y$) be correlated with $Y$.  Note that the punctuation example is only an extreme toy example. Experiments in Fig.5(a) show empirical evidence for this claim. The orange line corresponds to a randomly initialized generator (not trained to recognize any semantics), but the predictor can still make correct predictions with the random selected rationales.
> For the above reasons, reducing the capacity of the generator is not the primary focus  of our efforts. Nevertheless, we have also ever considered some of the above methods.
> **A2.1 (about reducing the capacity of the generator).**
> We do not deny that some methods can work. But it is non-trivial to design a good method that works well. Since we need to maintain the generotor's power for identifing the sentiment rationales, we need to be very careful when reducing the capacity of the generator to get a trade-off. Practically, we have ever considered to add spectral normalization (SN) to the generator's linear layer to restrict its Lipschitz continuity and produce a more flatter model. But we found that this trick actually hurts the model. When the SN is added, the model cannot be trained at all.
> **A2.2 (about adding regularization to the generator).**
> In fact, a previous method 3PLAYER [A] has tried to do this. It takes the unselected part into consideration and the regularization term for the generator is to make the unselected part to be unable be classified correctly. But it has been empirically verified by several later papers [B,C] that it does not improve the rationale quality significantly as compared to the vanilla RNP.
> **A2.3 (about training more steps for the predictor).**
> In the paper of FR, it has been empirically shown (Fig.1 in FR) that training the predictor with a higher learning rate than the generator (which is somewhat similar to training the predictor for more steps) unexpectedly results in a decrease in the quality of rationales. Instead, training the generator with a higher learning rate improves the rationale quality.
> **A2.4 (mode collapse).**
> Although we call the generator a generator, but actually in fact a selector which outputs binary masks (e.g., "0,1,1,0,0,1") instead of real text. How to measure the diversity is not easy. Also, the problem is not completely the same as mode collapse. A well-known method to address mode collapse in GANs is using a smaller learning rate for the generator [D]. But according to A2.3, this method does not work for the cooperative game of rationalization.
>
> We are grateful for your valuable suggestions, and we do not claim that this method is the sole or optimal solution to this problem. However, exploring more advanced methods is somewhat beyond the scope of this paper. We would like to respectfully clarify that our contributions in this paper are not limited to the proposed method. Another important contribution is the identification of this special sampling bias problem (**Q1** and **A1** above).

---

> ### Author Response · Authors · 2023-11-18
> **Edge cases, singular sentiment assumption, and the mathematical formulation.**
>
> **Q3.** There are a few edge cases which I am not convinced that A2I will be able to fix. Primarily, these deal with the circumstance where, in the toy example, the do not appear in the negative samples, and vice versa -- so is a token that appears almost exclusively in positive examples, and is a token that appears almost exclusively in negative examples. Such spurious correlations have been found in natural language tasks [3-4]. In these cases, it seems like the attacker would not be able to choose the corresponding token, and would thus still output random noise.
>
> **A3.** If we understand correctly, you are discussing the spurious correlations that already exist in the original dataset. It involves another line of research that is orthogonal to this paper. Existing research on causality has primarily focused on spurious correlations inherent in the dataset. However, our research investigates a novel question: if the dataset itself is clean and lacks spurious correlations, could the selection process of the generator introduce additional spurious correlations? This poses a unique research question in our study.
>
> Our research problem in this study is the spurious correlations introduced by the generator's selection (as implied in Section 4.2, trivial patterns in this paper refer to those independent with $Y$).
> We acknowledge that we cannot deal with those spurious correlations that exist in the original dataset. We see a recent study MCD [E] in the field of rationalization has already noticed this problem and addressed it well. As we have implied in our submission, our A2I is model-agnostic, we will consider building our A2I on top of MCD in the future, just like FR+A2I in our paper.
>
>
>
> **Q4.** Another concern deals with the singular sentiment assumption. This seems like a strong assumption that is very dataset and task specific, and the authors already discuss its failure modes in the appendices. The presence of negation seems to be another case where the assumption would be violated. As such, I am not convinced in the generalizability of the method to other datasets and tasks. Regardless, the authors should formulate this assumption mathematically in the text.
>
> **A4.**
> $S\in \{s_+,s_-\}$ constitutes indicative features for the category (e.g., sentiment orientation), i.e., $Y$ is primarily determined by $S$ within $X$. Additionally, there may exist descriptions $K\in \{k_+,k_-\}$ in $X$ related to sentiment but not decisive for the label (e.g., softened tones).
> Formally, the  singular sentiment assumption is $P(\phi(x,s_-)|Y=1)=0, P(\phi(x,s_+)|Y=0)=0$.
> The reason we make this assumption is that we think that $s_+$ and $s_-$ will not appear in one $x$ simultaneously, otherwise we would not be able to determine to which category this $x$ belongs. And if $s_+$ and $k_-$ appear in one $x$, that $x$ still belongs to $Y=1$. So, $K$ is still a trivial pattern, and this is what we discussed in the appendix.
>
> If you still think this assumption is too strong, we can make it a relaxed one: $P(\phi(x,s_-)|Y=1)=\epsilon=P(\phi(x,s_+)|Y=0), \ s.t., \epsilon \ll P(\phi(x,s_-))$, which means that it is very hard for the attacker to find $s_-$ from $X$ labeled $1$. Based on it, our analysis in the last paragraph of Section 4.2 still makes sense.
>
>
>
> **Q5.** Overall, the clarity of the paper could be improved. Some of the formulation sections are hard to parse. For example, the authors formulate the problem as one of sampling bias, which makes sense intuitively. However, the mathematical formulation and causal graphs for this section don't follow the prior work in sampling bias [2].
>
> **A5.** Thank you for your suggestion. But could you please provide more details about the unclarity? Sampling bias can take various forms [F], and the issue we discuss may differ from that in reference [2]. The sampling bias we discuss in this study arises from approximating $P(Y|Z,g)$ as $P(Y|Z)$. And Equation (7) shows why such an approximation can introduce a bias. The format of a causal graph depends on the process it describes. In rationalization, there are two different data-generating processes (DGP): one is the normal DGP of the original dataset, and another is the DGP of the rationale selection. In our study, the causal graph (Fig.4) describes a small local of the generator's updating process (i.e., the second DGP mentioned above) rather than the DGP of the original dataset, as most causality research does  (since our research problem is the spurious correlations introduced by the generator, rather than the spurious correlations in the original dataset). Thus, it is not surprising that our causal graph differs from that of [2].

---

> ### Author Response · Authors · 2023-11-18
> **Experiments with Bert and GNNs.**
>
> **Q6.** Concerns about applying A2I to BERT and GNNs
> **A6.** Thanks for your suggestion. We have now re-run RNP and FR with the Bert encoder on the three aspects of BeerAdvocate. Although fine-tuning Bert is still very difficult, we find that using a smaller learning rate for the Bert encoder than for the linear layer works somewhat. So we use a learning rate of $1e-5$ for the Bert encoder and $1e-4$ for the linear layer. The maximum sequence length is 256 (this is enough for the Beer dataset). The batch size is 24.
>
> From the second table, we see that Bert\_RNP performs much worse than GRU\_RNP, so there must be some unknown issues prevent RNP to work well with Bert. The sampling bias problem studied in our paper may not be the only obstacle that prevents Bert_RNP to work well, so in most cases, A2I does not improve Bert_RNP a lot.
>
> But from the first table, we see that Bert_FR improves Bert_RNP a lot. So the obstacles introduced by the Bert encoder may have somewhat been overcomed by FR. Under this case, our A2I improves Bert_FR a lot (more than $5\%$ in 6 of 9 settings). We believe that future researchers will further explore harnessing the power of Bert within the RNP framework. At that point, we can also build upon subsequent methods to incorporate our A2I.
>
> Notes: Due to limited GPU resources and time, we run the experiments with a fixed random seed 12252018 (inherited from the code provided by our baseline FR). Although not very solid, having the same seed for the nine different settings (3 different aspects $*$ 3 different sparsities) may somewhat reflect the stability of the effectiveness of A2I. We highlight the results only when FR+A2I gives an improvement over $5\%$ compared to FR.
>
> | Bert\_FR |  | Appearance |  |  |  |  |  | Aroma |  |  |  |  |  | Palate |  |  |  |  |
> |---:|:---:|:---:|---:|---:|---:|---:|---:|:---:|---:|---:|---:|---:|---:|:---:|---:|---:|---:|---:|
> |  |  | s | acc | p | r | f1 |  | s | acc | p | r | f1 |  | s | acc | p | r | f1 |
> | $S\approx 10\%$ | FR | 9.9 | 80.2 | 72.5 | 39.0 | 50.8 |  | 11.7 | 80.6 | 56.7 | 41.7 | 48.1 |  | 10.0 | 81.6 | 27.4 | 21.8 | 24.3 |
> |  | FR+A2I | 10.0 | 85.0 | 91.5 | 49.8 | **64.5** |  | 10.2 | 82.3 | 75.9 | 49.0 | **59.5** |  | 9.1 | 83.7 | 35.1 | 25.3 | **29.4** |
> |  |  |  |  |  |  |  |  |  |  |  |  |  |  |  |  |  |  |  |
> | $S\approx 20\%$ | FR | 19.6 | 84.8 | 56.6 | 60.5 | 58.5|  | 19.5 | 83.2 | 43.6 | 53.5 | 48.1 |  | 19.4 | 84.4 | 39.2 | 60.3 | 47.5 |
> |  | FR+A2I | 17.2 | 85.7 | 72.7 | 68.2 | **70.3** |  | 20.2 | 88.7 | 57.6 | 73.3 | **64.5** |  | 20.0 | 85.8 | 42.3 | 66.9 | 51.9 |
> |  |  |  |  |  |  |  |  |  |  |  |  |  |  |  |  |  |  |  |
> | $S\approx 30\%$ | FR | 29.9 | 86.1 | 51.5 | 84.0 | 63.9 |  | 28.7 | 81.6 | 18.1 | 32.7 | 23.3 |  | 28.6 | 82.6 | 12.4 | 28.0 | 17.2 |
> |  | FR+A2I | 30.5 | 87.0 | 51.9 | 86.4 | 64.8 |  | 30.2 | 84.8 | 39.7 | 75.7 | **52.1** |  | 29.0 | 83.2 | 13.1 | 30.0 | 18.2 |
>
> | Bert\_RNP |  | Appearance |  |  |  |  |  | Aroma |  |  |  |  |  | Palate |  |  |  |  |
> |---:|:---:|:---:|---:|---:|---:|---:|---:|:---:|---:|---:|---:|---:|---:|:---:|---:|---:|---:|---:|
> |  |  | s | acc | p | r | f1 |  | s | acc | p | r | f1 |  | s | acc | p | r | f1 |
> | $S\approx 10\%$ | RNP | 10.6 | 83.2 | 38.3 | 22.1 | 28.0 |  | 9.8 | 62.6 | 14.7 | 9.0 | 11.2 |  | 10.0 | 66.0 | 9.8 | 7.7 | 8.6 |
> |  | RNP+A2I | 10.2 | 85.4 | 46.8 | 26.0 | 33.5 |  | 10.4 | 77.5 | 15.0 | 9.8 | 11.9 |  | 10.5 | 76.2 | 10.1 | 8.4 | 9.2 |
> |  |  |  |  |  |  |  |  |  |  |  |  |  |  |  |  |  |  |  |
> | $S\approx 20\%$| RNP | 19.6 | 82.4 | 50.7 | 54.4 | 52.5 |  | 19.3 | 66.5 | 14.7 | 17.8 | 16.1 |  | 20.3 | 70.4 | 10.6 | 17.1 | 13.1 |
> |  | RNP+A2I | 19.8 | 84.5 | 55.6 | 60.2 | 57.8 |  | 19.3 | 78.1 | 15.7 | 19.0 | 17.2 |  | 19.1 | 75.4 | 10.8 | 16.3 | 13.0 |
> |  |  |  |  |  |  |  |  |  |  |  |  |  |  |  |  |  |  |  |
> |$S\approx 30\%$ | RNP | 29.2 | 82.2 | 23.2 | 37.0 | 28.5 |  | 29.6 | 76.3 | 15.8 | 29.5 | 20.6 |  | 29.4 | 72.1 | 11.1 | 25.8 | 15.5 |
> |  | RNP+A2I | 29.9 | 91.1 | 50.7 | 82.7 | 62.9 |  | 29.7 | 77.4 | 19.5 | 36.5 | 25.4 |  | 30.9 | 79.9 | 11.7 | 28.7 | 16.7 |
>
>
>
> Following your valuable suggestion, we have now also further applied RNP and our AI2 to explainable GNNs. We use a very widely used graph classification dataset in the field of explainable GNNs: BA2Motifs. There are labels for the gold rationales: the house motif for class 0 and the cycle motif for class 1. We report the overlap (F1 score) of the selected nodes and gold rationales. The base model is a 2-layer GCN. The results are as follows:
>
> | BA2Motifs | S | Acc | P | R | F1 |
> |:---:|:---:|:---:|:---:|:---:|---|
> | RNP | 20.3(2.5) | 95.2(1.9) | 36.5(5.5) | 36.5(2.2) | 36.4(3.8) |
> | RNP+A2I | 20.5(2.3) | 95.2(1.5) | 39.7(3.5) | 40.5(2.9) | **40.0(2.5)** |
> |  |  |  |  |  |  |
> | FR | 20.5(2.3) | 96.4(1.8) | 39.3(5.9) | 40.0(4.9) | 39.6(5.2) |
> | FR+A2I | 20.2(1.5) | 96.5(1.4) | 42.1(2.8) | 42.5(4.0) | **42.3(3.0)** |
>
> Notes: The numbers in "()" are the standard deviations.

---

> ### Author Response · Authors · 2023-11-18
> **Experiments with different random seeds.**
>
> **Q7.** The authors do not show any confidence intervals for their results, so it is unclear whether performance gains are statistically significant. They also only evaluate on two datasets, though these seem to be standard datasets in the RNP community.
>
> **A7.1. (standard deviation)** We now report the standard deviation. In our original experiments, we use a fixed random seed of 12252018 (inherited from the code provided by our baseline FR), because we think that experiments with 12 different settings (beer: 3 aspects $*$ 3 sparsity, hotel: 3 aspects) under the same random seed are somewhat sufficient to verify the stability of the models. Here we rerun RNP and RNP+A2I with 4 additional seeds and report the standard deviation over the five random seeds. Notes: The format of the numbers is "avg(std)".
>
> |  |  | Appearance |  |  |  |  |  | Aroma |  |  |  |  |  | Palate |  |  |  |  |
> |---:|:---:|:---:|---:|---:|---:|---:|---:|:---:|---:|---:|---:|---:|---:|:---:|---:|---:|---:|---:|
> |  |  | s | acc | p | r | f1 |  | s | acc | p | r | f1 |  | s | acc | p | r | f1 |
> | $S\approx 10\%$  | RNP | 9.0(1.2) | 81.5(1.5) | 83.4(8.0) | 40.3(5.5) | 54.2(5.9) |  | 9.2(1.2) | 83.7(1.7) | 84.1(2.7) | 49.7(5.5) | 62.3(4.1) |  | 9.7(0.3) | 83.2(1.8) | 69.1(2.1) | 53.8(1.9) | 60.5(1.8) |
> |  | RNP+A2I | 10.0(0.6) | 82.6(1.3) | 82.2(3.6) | 44.8(1.0) | 58.1(0.7) |  | 9.9(0.3) | 83.8(2.0) | 84.5(1.0) | 53.9(1.2) | 65.8(0.9) |  | 10.1(0.5) | 85.4(1.1) | 69.3(2.3) | 56.3(1.6) | 62.1(1.3) |
> |  |  |  |  |  |  |  |  |  |  |  |  |  |  |  |  |  |  |  |
> | $S\approx 20\%$  | RNP | 19.5(0.3) | 83.3(1.4) | 69.2(2.6) | 73.1(3.7) | 71.1(3.1) |  | 21.0(0.7) | 85.8(1.2) | 43.9(2.7) | 59.2(2.0) | 50.4(2.5) |  | 19.2(0.8) | 85.3(1.9) | 47.0(2.0) | 72.7(5.0) | 57.0(2.9) |
> |  | RNP+A2I | 20.0(0.1) | 85.2(2.6) | 72.6(0.9) | 78.6(1.0) | 75.5(1.0) |  | 19.5(0.3) | 86.3(1.7) | 50.1(1.0) | 62.7(1.4) | 55.6(1.1) |  | 18.8(0.8) | 86.2(0.6) | 48.4(3.0) | 72.8(2.9) | 58.2(2.9) |
> |  |  |  |  |  |  |  |  |  |  |  |  |  |  |  |  |  |  |  |
> | $S\approx 30\%$  | RNP | 30.5(1.1) | 85.5(2.4) | 55.9(2.6) | 92.2(3.2) | 69.6(2.7) |  | 31.2(0.4) | 85.9(3.7) | 39.2(1.8) | 78.5(3.1) | 52.3(2.3) |  | 29.0(0.9) | 86.1(2.6) | 31.1(2.7) | 72.6(6.1) | 43.6(3.7) |
> |  | RNP+A2I | 29.7(0.2) | 85.7(1.7) | 59.5(0.7) | 95.5(1.3) | 73.4(0.9) |  | 29.4(1.1) | 88.0(1.2) | 44.8(1.6) | 84.6(3.7) | 58.5(2.0) |  | 28.5(0.9) | 86.4(0.8) | 32.5(1.7) | 74.7(5.0) | 45.3(2.5) |
>
>
> Due to the large amount of GPU resources consumed by Bert related experiments, the random seed experiments for FR are still in preparation.
>
> **A7.2. (datasets)**
> We fully understand your concern. However, datasets containing manually annotated rationales are precious, and it is common to utilize only two datasets in the field of rationalization. Our baseline method FR, and two other relevant papers [C,E] all use two datasets.
> Although we use only two datasets, each of these datasets contains three independently annotated aspects and each aspect is trained independently. Thus, to some extent, they can be considered as six datasets to some extent.
>
>
> References
> [A] Rethinking Cooperative Rationalization: Introspective Extraction and Complement Control. EMNLP 2019.
> [B] Invariant Rationalization. ICML 2020.
> [C] Understanding Interlocking Dynamics of Cooperative Rationalization. NeurIPS 2021.
> [D] GANs Trained by a Two Time-Scale Update Rule Converge to a Local Nash Equilibrium. NeurIPS 2017.
> [E] D-Separation for Causal Self-Explanation. NeurIPS 2023.
> [F] A Survey on Bias and Fairness in Machine Learning. ACM computing surveys 2021.

---

### Author Response · Authors · 2023-11-18
**General response**

My sincere apologies for the delayed response. The experiments with BERT took longer than anticipated. Thank you for your patience and understanding.

Here is a summary of the general reponses:

1. We have uploaded the code. The code and instructions are at: https://anonymous.4open.science/r/A2I-6612.

2. We have reported the results with BERT encoder (bert-base-uncased) (to Reviewer TefW, MTBk, and LkPz). Code: https://anonymous.4open.science/r/BERT-A2I-56AA/.

3. We have reported the results of experiments conducted with five different random seeds (to Reviewer TefW, and QZhC).

4. We have further applied RNP, FR, and our A2I to a graph classification dataset (to Reviewer TefW, MTBk, QZhC, and LkPz).

5. We have clarified some specific misunderstandings.

---

### Public Comment · ~Yu_Fan1 · 2023-11-20
**Questions for A2I**

Hello, this is a very interesting work! I've also been researching the interpretability lately. But I still have some questions about the experiments of the thesis and would like to get your answers.

(1) In the related work, you mentioned that 3PLAYER, DME and A2R are all very relevant to your research work, so why don't you use these work as baselines?

(2) I find that you used RNP, Inter_RAT and FR as baselines, why in Table 1 and Table 2, only RNP+A2I and FR+A2I but not Inter_RAT+A2I?

(3) Recently there are many cutting edge interpretable work related to your research work, I hope you can consider them as baselines, such as [1][2].

Looking forward to your response!

References

1. Decoupled Rationalization with Asymmetric Learning Rates: A Flexible Lipschitz Restraint.  KDD 2023

2. MGR: Multi-generator Based Rationalization. ACL 2023

---

> ### Author Response · Authors · 2023-11-20
> **Thank you for your interest in this study.**
>
> **A1.** The experiments are designed for two purposes:
> 1. To verify the claims proposed in this paper. We compare with the vanilla RNP to verify the existence of the sampling bias problem (this part is also verified by Section 5.1) and the effectiveness of our AI2 in addressing it. And we think this is the most important point of this study.
> 2. To show the competitiveness of the proposed method.  So we compare with two recent methods FR and Inter\_RAT to support this point.
>
> All three methods you mentioned are somewhat old in the current literature. 3PLAYER was published in 2019. DMR and A2R were published in 2021. Besides, the performance of them is not as good as the more recent methods we have compared with. Experiments in A2R show that 3PLAYER sometimes does not even beat the vanilla RNP (note that A2R and 3PLAYER are published by the same team of authors). Also, FR has been empirically shown to outperform DMR and A2R by a large margin (up to $10.3\%$ in terms of F1 score). Note that the datasets we use in this study are the same as those in FR.
>
> So, comparing with these old methods can neither support the first nor the second goal we mentioned above, and thus can bring few new insights to the readers. Moreover, these methods involve extra modules and hyperparameters. If you have ever reimplemented these methods yourself, you will know that it is not easy to find good parameters for them, which may lead to unfair comparisons.
>
> **A2.** There are three reasons that may address your question.
> 1. Till our submission, Inter\_RAT has not been formally published. There are no instructions in its released code. And the implementation details are not very clear. So, building our A2I on top of it is hard.
>
> 2. The goal of the comparison with advanced methods is to show the competitiveness of the proposed method. From the results, it seems that FR is much better than Inter\_RAT. So, comparing FR and FR+A2I can better support this goal.
>
> 3. RNP+A2I has already outperformed Inter\_RAT by a large margin, showing that A2I is a better method than Inter\_RAT in this scenario.
>
>
> **A3.** Thank you for providing the new references. They are relevant papers, but they are too recent to compare.
> KDD 2023: August 6-10, 2023.
> ACL 2023: July 9-14, 2023.
> Submission ddl for ICLR 2024: September 28, 2023.
>
> According to the ICLR policy (https://iclr.cc/Conferences/2024/ReviewerGuide), authors are not required to compare with papers published within the last four months.
>
> Nevertheless, we are grateful for your suggestions. We find that the core idea of the DR paper [1] provides a very simple and flexible trick. It is orthogonal to our research and can be easily combined with our A2I. We are preparing the experiments for DR+A2I.
>
> We will release the code and instructions (they are now visible only to reviewers) in case of acceptance.
>
> We welcome any further questions! Just add your comments below.
>
> [1] Decoupled Rationalization with Asymmetric Learning Rates: A Flexible Lipschitz Restraint. KDD 2023
>
> update:
> We have done a quick test for DR on part of the Beer dataset. The core idea of DR is to assign $\lambda=\frac{\eta_g}{\eta_p}>1$ for RNP, where $\eta_g, \eta_p$ are the learning rates for the generator and the predictor, respectively. In the original paper, the authors provided a heuristic method to relate $\lambda$ and sparsity. For simplicity in reimplementation, we simply set $\lambda$ to 10, as there is some empirical evidence from the original paper suggesting that 10 is a reasonable choice. The results are as follows:
>
> |  | Appearance |  |  |  |  |  | Aroma |  |  |  |  |  | Palate |  |  |  |  |
> |---|---|---|---|---|---|---|---|---|---|---|---|---|---|---|---|---|---|
> |  | s | acc | p | r | f1 |  | s | acc | p | r | f1 |  | s | acc | p | r | f1 |
> | DR($\lambda=10$) | 19.9 | 81.5 | 79.0 | 85.0 | 81.9 |  | 19.2 | 84.2 | 63.2 | 78.0 | 69.8 |  | 19.1 | 87.1 | 51.1 | 78.6 | 61.9 |
> | DR+A2I | 19.4 | 82.4 | 82.1 | 85.9 | 83.9 |  | 19.3 | 85.0 | 65.8 | 81.3 | 72.7 |  | 18.9 | 88.0 | 52.4 | 79.6 | 63.2 |
>
> Finally,  we reiterate that the research questions addressed in these new works are orthogonal to ours, and there is potential to integrate different methods to address distinct issues.

---

> > ### Public Comment · ~Yu_Fan1 · 2023-11-21
> > **Thank you for your response**
> >
> > Thank you for your response!
> > Looking forward to further interaction with you when the ICLR results are out.